# Distribution-Aware Multi-Granularity Phase Coding: Towards Lower Conversion Error for Spike-Driven Large Language Models

**Hanyuan Zheng**[1*], **Haozhen Zhang**[1*], **Tianshuo Chen**[1*], **Zhaogeng Liu**[1*], **Yi Chang**[1,2,3†], **Bin Gu**[1†]

School of Artificial Intelligence, Jilin University[1], International Center of Future Science, Jilin University[2]
Engineering Research Center of Knowledge-Driven Human-Machine Intelligence, MOE[3]
`{hyzheng25,hzzhang23,tschen25,zgliu20}@mails.jlu.edu.cn,`
`yichang@jlu.edu.cn, jsgubin@gmail.com`

## Abstract

Spiking large language models (LLMs) offer significant advantages on neuromorphic hardware, yet training them from scratch remains prohibitively expensive. A promising alternative is ANN-to-SNN conversion, which reuses pretrained ANN weights while minimizing conversion error. However, existing conversion frameworks neglect activation distributions, as reflected in SNN neurons with rate or temporal coding to map uniformly distributed rather than distribution-aligned discrete values, thus causing latent conversion error arising from distribution misalignment. To tackle this problem, we propose a distribution-aware multi-granularity phase coding approach, which achieves reasonable discrete value allocation by minimizing conversion error relative to activation distributions. Specifically, multi-granularity phase coding extends conventional phase coding with multiple learnable bases, incorporating representational capacity across different granularities. Building on this coding scheme, we further propose a novel ANN-to-SNN conversion paradigm designed towards lower conversion error. In particular, our paradigm utilizes the activation distributions of hidden layers to sample data for cost-efficient neuron training, without requiring fine-tuning of model weights. Theoretically, we provide a convergence guarantee for the neuron training algorithm. Extensive experiments on the LLaMA model confirm the effectiveness of both our coding scheme and conversion paradigm. Concretely, our spiking LLM attains the lowest perplexity with ANN-level accuracy, accompanied by a 42% reduction in energy consumption of MAC and AC operations. Our code is available at `https://github.com/njzhenghy/SpikingLLM`.

## 1 Introduction

Large language models (LLMs), exemplified by GPT-4 (Achiam et al., 2023), Qwen3 (Yang et al., 2025), and LLaMA3 (Dubey et al., 2024), achieve remarkable performance across a wide range of natural language processing tasks through training on massive text corpora. However, the transformer architecture in LLMs relies on dense matrix multiplications, where intensive Floating-Point Multiplication and Addition (MAC) operations result in prohibitive energy consumption (Vaswani et al., 2017; De Vries, 2023). This substantial challenge during training and inference necessitates the pursuit of energy-efficient paradigms for LLMs. In contrast to conventional neural networks, spiking neural networks (SNNs) have received increasing attention due to their energy efficiency in mimicking biological neurons, thereby offering a promising solution. Building on this foundation, spiking LLMs have recently been developed, showing promise for efficient execution on neuromorphic hardware (Xing et al., 2025; Chen et al., 2025a; Zhengzheng & Zhu, 2025).

Research on SNNs has primarily focused on two approaches: direct training and ANN-to-SNN conversion. Direct training methods (Neftci et al., 2019; Zenke & Vogels, 2021; Lee et al., 2016)

---

*These authors contributed equally as co-first authors and are listed in random order.
†Co-corresponding authors.

typically adopt surrogate gradients during backpropagation to address the non-differentiability of spiking neurons. Nevertheless, training SNNs from scratch is prohibitively costly in both time and resources, particularly at the scale of LLM parameters. In contrast, ANN-to-SNN conversion (Tong et al., 2022; Hao et al., 2023a; Yang et al., 2022; Chen et al., 2025b) offers a more efficient paradigm, typically reusing pretrained ANN weights in the spiking model while minimizing conversion error to achieve effective conversion. Since minimizing conversion error is often less costly than direct training, ANN-to-SNN conversion demonstrates greater generality in resource-constrained environments (Ding et al., 2021).

Unfortunately, there exists the conversion error arising from distribution misalignment, which is a long-standing inherent problem in such conversions (Datta & Beerel, 2022). However, current ANN-to-SNN conversion frameworks for LLM tend to overlook non-uniform activation distributions, leading to latent errors owing to distributional misalignment (Chen et al., 2025b). As shown in Figure 1, activations within a single layer are generally non-uniformly distributed, and activation distributions differ across layers. Unfortunately, in the coding schemes of existing spiking LLMs, rate (Wu et al., 2019; Sengupta et al., 2019) or temporal coding methods (Mostafa, 2017; Zhao et al., 2025) typically map discrete values by discretizing activation values into uniformly partitioned intervals, rather than aligning with the large-scale non-uniform activation distributions observed in practice. Furthermore, distinct activation distributions across different components of large models pose an additional challenge, highlighting the need for a learnable and adaptive framework capable of handling heterogeneous distributions (Zhang et al., 2018).

To address the challenge mentioned above, we introduce an alternative coding scheme referred to as phase coding (Kim et al., 2018; Zhang et al., 2020) and significantly enhance it by proposing distribution-aware multi-granularity phase coding. Conventional phase coding can realize non-uniform allocation of mapped discrete values by adjusting the base. Building on this observation, the proposed distribution-aware multi-granularity phase coding integrates representational capacities at different granularities through multiple learnable bases, thereby offering enhanced flexibility in discrete value allocation. The final outcome is that it can achieve a more reasonable discrete value allocation by minimizing distribution-related conversion errors, which is essential for ensuring the performance of spiking LLMs after conversion.

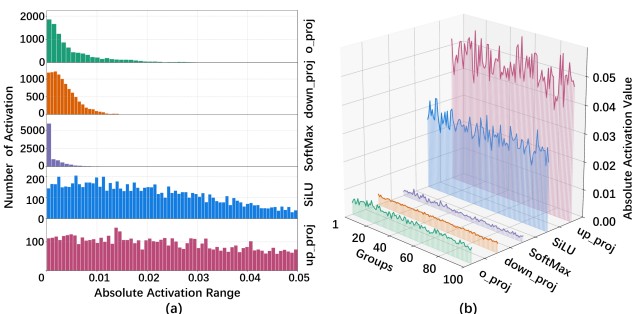

Figure 1: Distributions of the absolute activation values across layers in a large language model. (a) The activation values of each layer are truncated between 0 and 0.05, and the histogram reveals the uneven distribution of activation values. (b) The activation values are divided into 100 groups, with each group representing the average, showing the uneven distribution trend of activation values.

Furthermore, we develop a novel ANN-to-SNN conversion paradigm built upon the aforementioned coding scheme. The central component of the paradigm is a cost-efficient alternating optimization neuron training algorithm, designed to minimize conversion error relative to activation distributions. Specifically, we tune only the neuron parameters using data pre-sampled from the corresponding hidden-layer activation distributions, which eliminates the forward and backward propagation of network layers and renders our paradigm highly cost-efficient in conversion. In summary, our paradigm yields spiking LLMs with both low conversion error and a highly cost-efficient conversion under a convergence guarantee.

Our contributions are summarized as follows:

- **Multi-granularity Phase Coding.** We propose a distribution-aware, multi-granularity phase coding scheme with multiple learnable bases, which enables flexible and adaptive allocation of discrete value mappings.

- **Distribution-Aware Conversion Paradigm.** We establish a distribution-aware paradigm that breaks the uniform discretization of rate and temporal coding, facilitating faithful ANN-to-SNN conversion.

- **Theoretical Convergence Guarantee.** We analyze the convergence for the proposed alternating optimization neuron training algorithm, based on the gap between the objective function before and after smoothing.

- **A remarkable Spike-Driven LLM.** Our Spiking LLM achieves the lowest perplexity while preserving ANN-level accuracy, setting new state-of-the-art results, meanwhile reducing energy consumption of MAC and AC operations by 42.0% compared to its ANN counterpart.

## 2 RELATED WORKS

### 2.1 ANN-TO-SNN CONVERSION

Existing ANN-to-SNN conversion methods are primarily divided into one-stage and two-stage approaches (Chen et al., 2025a). The former involves not performing any further optimization on the converted SNN and directly converting the ANN to an SNN model. This approach is commonly used when the target ANN is built upon ReLU functions, as the output of ReLU can be effectively approximated by the firing rate of spiking neurons (Cao et al., 2015). Building on the insight discussed above, both Diehl et al. (2015) and Sengupta et al. (2019) employ normalization techniques to further improve conversion performance. Additionally, Bu et al. (2023) propose, from a theoretical perspective, the use of QCFS functions to replace ReLU functions in order to effectively reduce conversion error. The two-stage approach, on the other hand, focuses on optimizing the converted SNN to ensure its performance. Hao et al. (2023a) classify the unevenness error into four cases and propose an optimization strategy based on residual membrane potential to reduce error. Hao et al. (2023b) focus on addressing the conversion error caused by one additional (or one less) spike by shifting the initial membrane potential. Chen et al. (2025a) adopt a coarse-to-fine calibration optimization strategy to optimize the converted SNN. However, these approaches either struggle to scale to transformer-based LLMs or still incur high optimization costs for the converted spiking LLMs.

### 2.2 SPIKING LLM

Spiking LLMs, noted for their low energy consumption, are gradually emerging as a promising direction in the field of large-scale models. Despite this promise, research on spiking LLMs remains limited. Early efforts include SpikingBERT (Bal & Sengupta, 2024), which leverages the average spiking rate of neurons at equilibrium and incorporates knowledge distillation to enhance both training efficiency and model performance. SpikeGPT (Zhu et al., 2023) adapts RWKV by combining spiking activations with sequential attention, demonstrating that autoregressive language generation is feasible within the spiking paradigm. More recently, SpikeLLM (Xing et al., 2025) has introduced a hybrid co-architecture that integrates SNNs with quantized ANNs, scaling spiking models to the billion-parameter regime (7–70B) and achieving improved energy efficiency. FAS (Chen et al., 2025a) enables the conversion of pretrained ANN-based LLMs into spiking counterparts through a two-stage calibration, resulting in lower energy consumption and latency. However, existing Spiking LLM frameworks typically rely on uniform rate coding, which overlooks the non-uniform distribution of activations and consequently introduces latent conversion error.

## 3 PRELIMINARY

Spiking coding is a scheme that determines how continuous values are encoded into a sequence of spikes, with rate coding, temporal coding, and phase coding being the most widely used schemes. In particular, phase coding combines the characteristics of temporal coding and rate coding, achieving a higher representational density than other coding schemes under the same total timestep $T$.

Specifically, similar to temporal coding, where spikes produced at different timesteps $t$ represent different values, phase coding assigns distinct weights to each $t$ within the total timestep $T$. Unlike the uniform phase values induced by the typically employed linear proportion $\frac{T-t}{T}$ in temporal coding (Rueckauer & Liu, 2018; Han & Roy, 2020), phase coding assigns each timestep $t$ a phase value $B^{-t}$, where $B$ is the base of the phase. At the same time, it preserves the multi-spike representation inherent in rate coding, thereby enhancing the representational capacity within a finite total $T$. By

combining these advantages, phase coding achieves an expansion of the number of encoded discrete values to $2^T$, which reduces the total number of timesteps $T$ in an exponential manner.

The biological manifestation of generic phase coding has been demonstrated by (Montemurro et al., 2008) and is further advanced in ANN-to-SNN conversion (Hwang & Kung, 2024; Hwang et al., 2024). Its corresponding neuron dynamic procedure is characterized by the threshold $\theta(t)$, reset strength $h(t)$, and output weight $d(t)$, as detailed in the following equation:

$$v(1) = \sum_{t^{pre}=1}^{T} I(t^{pre}), \ v(t+1) = v(t) - h(t)s(t), \ O(t) = d(t)s(t), \tag{1}$$

$$s(t) = \Theta\big(v(t)-\theta(t)\big) = \Theta\big(v(1) - \sum_{j=1}^{t-1} h(j)s(j) - \theta(t)\big), \tag{2}$$

where $\Theta(\cdot)$ denotes the Heaviside step function and the initial membrane potential $v(1)$ receives the output $I(t^{pre})$ from the pre-layer. For timestep $t \in \{1, 2, \ldots, T\}$ of the current neuron, $v(t)$ denotes the membrane potential, $s(t) \in \{0, 1\}$ denotes the binary spike, $O(t)$ denotes the output signal. Typically, for the conventional phase coding, $h(t)$, $d(t)$, and $\theta(t)$ are specified as $B^{-t}$ as follows:

$$v(t+1) = v(t) - B^{-t}s(t), \ O(t) = B^{-t}s(t), \ s(t) = \Theta\big(v(t)-B^{-t}\big). \tag{3}$$

Notably, the activation value can be approximated within a limited timestep $T$.

## 4    METHODOLOGY

In this section, we first explain our motivation from the perspective of information theory. Next, we introduce our multi-granularity phase coding. Finally, we elaborate on the proposed distribution-aware ANN-to-SNN conversion paradigm, as shown in Figure 2.

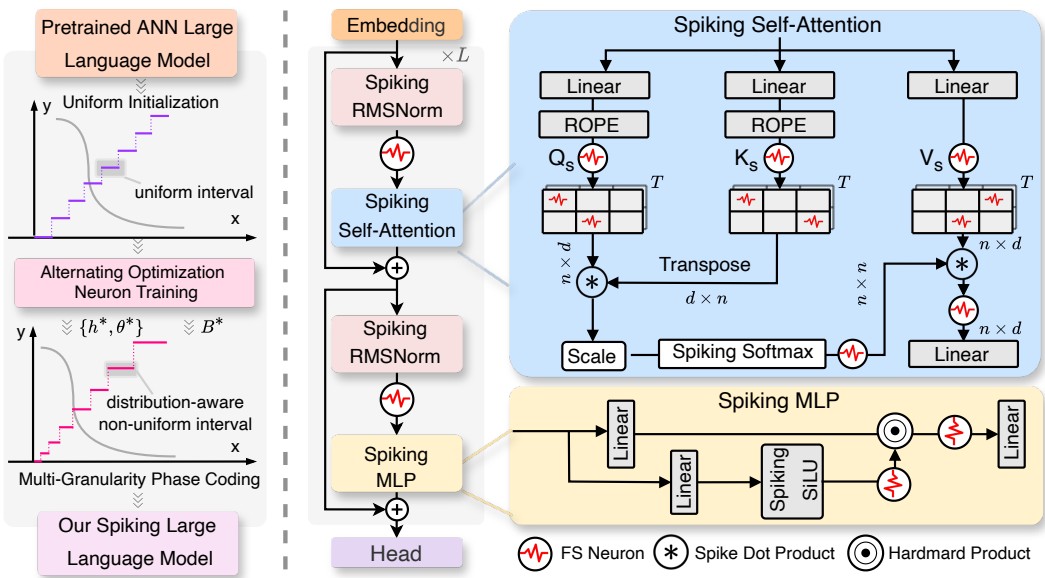

Figure 2: ANN-to-SNN conversion paradigm based on multi-granularity phase coding. Left: conversion pipeline with alternating neuron optimization. Right: spiking LLM built on SNN neurons.

### 4.1    MOTIVATION FROM INFORMATION THEORY

Considering the distribution of activation values, the ANN-to-SNN conversion error can be characterized by the mean squared error as follows:

$$E = \int p(x)\big(\hat{x} - x\big)^2 dx, \tag{4}$$

where $\hat{x}$ denotes the SNN neuron's approximation of the ANN activation value $x$. The conversion error can be regarded as equivalent to the quantization distortion in information theory. From the information-theoretic perspective, SNN coding is analogous to quantization in its allocation of discrete values. When an SNN employs $M$ allocation intervals, $\lambda(x)$ represents the relative density of these intervals. Consequently, the conversion error is equivalent to the quantization distortion, as formalized in Theorem 1.

**Theorem 1** (cf., (Gray & Neuhoff, 2002)). *For an arbitrary quantizer $q$, the asymptotic average distortion with $M$ quantization intervals can be expressed by rewriting Bennett's integral in terms of the point density function:*

$$D(q) = \int p(x)\big(x - q(x)\big)^2 dx \simeq \frac{1}{12}\frac{1}{M^2}\int \frac{p(x)}{\lambda^2(x)}dx, \tag{5}$$

*where $p(x)$ denotes the probability density function (PDF) of the input signal, and $\lambda(x)$ denotes the point density function of the quantizer.*

To minimize the ANN-to-SNN conversion error, Corollary 1 specifies the optimal allocation principle, namely assigning larger $\lambda(x)$ to regions with higher probability density.

**Corollary 1.** *Let $D(q)$ be the asymptotic distortion in Theorem 1. The point density function $\lambda(x)$ that minimizes $D(q)$, subject to the normalization constraint $\int \lambda(x)dx = 1$, is given by:*

$$\lambda^*(x) = \frac{[p(x)]^{1/3}}{\int [p(u)]^{1/3}du} \Rightarrow \lambda^*(x) \propto [p(x)]^{1/3}. \tag{6}$$

In LLMs, the activation distribution $p(x)$ is inherently non-uniform. Consequently, the optimal allocation function $\lambda^*(x)$ also exhibits non-uniformity. This implies the necessity of a distribution-aware coding strategy in SNNs, whereby regions of higher activation density are allocated more quantization intervals, while regions of lower density receive fewer intervals.

## 4.2 MULTI-GRANULARITY PHASE CODING

Motivated by this, we introduce multi-granularity phase coding, which adaptively allocates bases $\{B_1, B_2, \ldots, B_n\}$ of different granularities to non-uniform activations within a small timestep $T$. In particular, the alteration of phase values introduced by our multi-granularity phase coding, relative to conventional phase coding, is formally defined as follows:

$$\{B^{-t}\}_{t=1}^T \to \{B_1^{-1}, B_1^{-2}, \ldots, B_2^{-t}, B_2^{-(t+1)}, \ldots, B_n^{-T}\}. \tag{7}$$

This design offers a more flexible discrete value allocation, effectively minimizing the expected conversion error $E$. A more intuitive illustration is provided in Figure 3. The non-uniform discrete value allocation introduced by multi-granularity phase coding allows us to align the mapped discrete values with the activation distribution by tuning the bases (Figure 3, left). Consequently, intervals with denser activations are allocated more discrete values rather than being uniformly distributed. The distinction in conversion error relative to uniform discrete value allocation is illustrated in Figure 3 (right), where the reduction in conversion error can be easily observed from the shaded area.

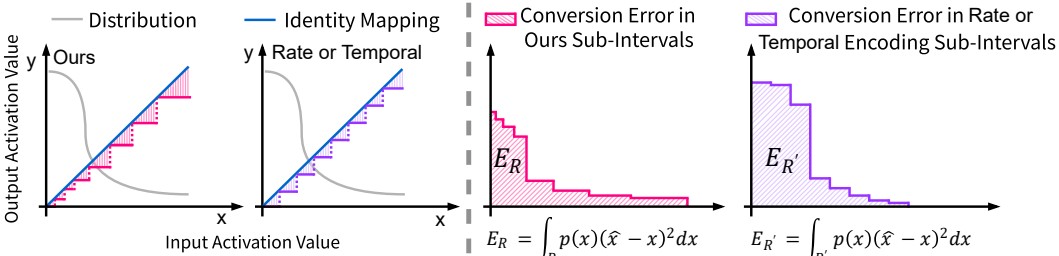

Figure 3: Comparison of discrete value allocation and conversion error between multi-granularity phase coding and rate or temporal coding. The shaded area $E_R$ and $E_{R'}$ represent conversion errors under different codings.

By incorporating the dynamics from Section 3, we can obtain the neuron dynamic procedure with multi-granularity phase coding, which is obtained by extending the conventional formulation in

Equation (3) and is formulated as follows:

$$v(t+1) = v(t) - h(t)s(t), \ O(t) = d(t)s(t), \ s(t) = \Theta\big(v(t) - \theta(t)\big),$$
$$\{d(t)\}_{t=1}^T = \{B_1^{-1}, B_1^{-2}, \ldots, B_2^{-t}, B_2^{-(t+1)}, \ldots, B_n^{-T}\}. \tag{8}$$

For clarity, we denote $\{h(t)\}_{t=1}^T$, $\{\theta(t)\}_{t=1}^T$, $\{d(t)\}_{t=1}^T$, $\{B_i\}_{i=1}^n$ as $\boldsymbol{h}$, $\boldsymbol{\theta}$, $\boldsymbol{d}$, and $\boldsymbol{B}$. In contrast to Equation (3), we remove the constraint $\theta(t) = h(t) = d(t)$ and instead treat $\{\boldsymbol{h}, \boldsymbol{\theta}\}$ as learnable parameters that are decoupled from $\boldsymbol{d}$. Equation (8) introduces our multi-granularity design, in which the neuron parameters are no longer constrained to a single base, but instead are constructed from multiple bases. This generalizes the conventional single-base scheme in Equation (3).

## 4.3 Distribution-Aware Conversion Paradigm

The proposed spiking LLM architecture is illustrated in Figure 2. By introducing SNN neurons with multi-granularity phase coding prior to linear layers and matrix operations, activation values are converted into spike signals, thereby avoiding floating-point matrix multiplications. For the SNN neuron with multi-granularity phase coding, our objective is to align its discrete value allocation with the activation distribution while minimizing conversion error, as introduced in Section 4.1. Toward this end, we consider the following expected conversion error:

$$\min_{\{\boldsymbol{h}, \boldsymbol{\theta}\}, \boldsymbol{B}} \int p(x)\big(SN(x; \{\boldsymbol{h}, \boldsymbol{\theta}\}, \boldsymbol{B}) - x\big)^2 dx, \tag{9}$$

where $p(x)$ is the PDF of activation $x$, and $SN(\cdot) = \sum_{t=1}^T O(t)$ is the discrete value obtained by mapping the activation value through the SNN neuron. In practice, for each neuron, the corresponding activation distribution can be estimated from a batch of input text and then downsampled to construct a training dataset $\boldsymbol{X}$ composed of activation samples. Ultimately, we formulate the target problem as an empirical conversion error minimization problem, as follows:

$$\min_{\{\boldsymbol{h}, \boldsymbol{\theta}\}, \boldsymbol{B}} \|SN(\boldsymbol{X}; \{\boldsymbol{h}, \boldsymbol{\theta}\}, \boldsymbol{B}) - \boldsymbol{X}\|^2. \tag{10}$$

To effectively solve the optimization problem formulated in Equation (10), we propose an alternating optimization neuron training algorithm, as presented in Algorithm 1. Specifically, we alternate between optimizing $\{\boldsymbol{h}, \boldsymbol{\theta}\}$ and $\boldsymbol{B}$. Due to the non-differentiability of the Heaviside step function, updates to $\{\boldsymbol{h}, \boldsymbol{\theta}\}$ are carried out using a sigmoid-based surrogate gradient (Wu et al., 2018). In the case of a fixed number of granularities, we address timestep allocation through an adaptive granularity allocation method, with full details provided in Appendix C. With Algorithm 1, model weights do not require fine-tuning, and the neuron training dataset is obtained through pre-sampling. Combined, these eliminate the need for forward and backward propagation through network layers and restrict propagation to neurons alone, rendering our training algorithm highly cost-efficient. Furthermore, the handling of other nonlinear operations in the model (e.g., RMSNorm, activation–activation multiplication, Softmax, and SiLU activation function) is provided in Appendix B.

## 5 Analysis

In this section, we first introduce several necessary assumptions and then present the convergence analysis results of the neuron training algorithm based on activation distributions.

## 5.1 Assumptions

We denote the original objective function $\|SN(\boldsymbol{X}; \boldsymbol{h}, \boldsymbol{\theta}, \boldsymbol{B}) - \boldsymbol{X}\|^2$ by $f(\boldsymbol{h}, \boldsymbol{\theta}, \boldsymbol{B})$, and the smoothed objective function by $g(\boldsymbol{h}, \boldsymbol{\theta}, \boldsymbol{B})$. In updating $\boldsymbol{h}$ and $\boldsymbol{\theta}$, surrogate gradients are employed to carry out gradient descent. This process can be regarded as gradient descent utilizing the true gradient of the smoothed objective function. Based on this perspective, we analyze the convergence behavior of the alternating optimization scheme. Since the parameters $\boldsymbol{h}$ and $\boldsymbol{\theta}$ are optimized simultaneously, we introduce the variable $\boldsymbol{o}$ to represent them collectively. A more detailed explanation of the symbols employed in the analysis is provided in Appendix A.

---

**Algorithm 1** Alternating Optimization Neuron Training Algorithm

---
1: **Input:** Training dataset $\boldsymbol{X}$, optimization steps $N$, $N_1$ and $N_2$, learning rate $\eta_1$ and $\eta_2$, neuron parameters $\{h(t), \theta(t)\}_{t=1}^T$ and $\{B_1^{-1}, B_1^{-2}, \ldots, B_2^{-t}, B_2^{-(t+1)}, \ldots, B_n^{-T}\}$.
2: Initialize $\{\boldsymbol{h}_0^{(0)}, \boldsymbol{\theta}_0^{(0)}\}, \boldsymbol{B}_0^0$.
3: **for** $i = 0, \cdots, N-1$ **do**
4:     **for** $j = 0, \cdots, N_1 - 1$ **do**
5:         Compute $\mathcal{L}_{MSE}(\boldsymbol{h}_i^{(j)}, \boldsymbol{\theta}_i^{(j)}; \boldsymbol{X}, \boldsymbol{B}_i^{(0)})$.
6:         Update $\{\boldsymbol{h}_i^{(j+1)}, \boldsymbol{\theta}_i^{(j+1)}\} = \{\boldsymbol{h}_i^{(j)}, \boldsymbol{\theta}_i^{(j)}\} - \eta_1 \hat{\nabla}_{\boldsymbol{h}, \boldsymbol{\theta}} \mathcal{L}_{MSE}(\boldsymbol{h}_i^{(j)}, \boldsymbol{\theta}_i^{(j)}; \boldsymbol{X}, \boldsymbol{B}_i^{(0)})$.
7:     **end for**
8:     Update $\{\boldsymbol{h}_{i+1}^{(0)}, \boldsymbol{\theta}_{i+1}^{(0)}\} = \{\boldsymbol{h}_i^{(N_1)}, \boldsymbol{\theta}_i^{(N_1)}\}$.
9:     **for** $j = 0, \cdots, N_2 - 1$ **do**
10:         Compute $\mathcal{L}_{MSE}(\boldsymbol{B}_i^{(j)}; \boldsymbol{X}, \boldsymbol{h}_{i+1}^{(0)}, \boldsymbol{\theta}_{i+1}^{(0)})$.
11:         Update $\boldsymbol{B}_i^{(j+1)} = \boldsymbol{B}_i^{(j)} - \eta_2 \nabla_{\boldsymbol{B}} \mathcal{L}_{MSE}(\boldsymbol{B}_i^{(j)}; \boldsymbol{X}, \boldsymbol{h}_{i+1}^{(0)}, \boldsymbol{\theta}_{i+1}^{(0)})$.
12:     **end for**
13:     Update $\boldsymbol{B}_{i+1}^{(0)} = \boldsymbol{B}_i^{(N_2)}$.
14: **end for**
15: **Output:** Neuron parameters $\{\boldsymbol{h}^*, \boldsymbol{\theta}^*\}, \boldsymbol{B}^*$.

---

**Assumption 1** (Lipschitz Gradient). *There exist constants $L_1, L_2 > 0$, for $\forall \mathbf{o}_1, \mathbf{o}_2$ and $\forall \mathbf{B}_1, \mathbf{B}_2$, such that:*

$$\|\nabla_{\boldsymbol{o}} g(\boldsymbol{o}_1, \boldsymbol{B}) - \nabla_{\boldsymbol{o}} g(\boldsymbol{o}_2, \boldsymbol{B})\| \leq L_1 \|\boldsymbol{o}_1 - \boldsymbol{o}_2\|, \tag{11}$$

$$\|\nabla_{\boldsymbol{B}} f(\boldsymbol{o}, \boldsymbol{B}_1) - \nabla_{\boldsymbol{B}} f(\boldsymbol{o}, \boldsymbol{B}_2)\| \leq L_2 \|\boldsymbol{B}_1 - \boldsymbol{B}_2\|. \tag{12}$$

**Assumption 2** (Polyak–Łojasiewicz (PL) condition). *There exist constants $\mu_1, \mu_2, \mu_3 > 0$ such that:*

$$\|\nabla g(\boldsymbol{o}, \boldsymbol{B})\|^2 \geq 2\mu_1 \big(g(\boldsymbol{o}, \boldsymbol{B}) - g^*\big), \tag{13}$$

$$\|\nabla_{\boldsymbol{o}} g(\boldsymbol{o}, \boldsymbol{B})\|^2 \geq 2\mu_2 \big(g(\boldsymbol{o}, \boldsymbol{B}) - g^{*(\boldsymbol{B})}(\boldsymbol{o}, \boldsymbol{B})\big), \tag{14}$$

$$\|\nabla_{\boldsymbol{B}} f(\boldsymbol{o}, \boldsymbol{B})\|^2 \geq 2\mu_3 \big(f(\boldsymbol{o}, \boldsymbol{B}) - f^{*(\boldsymbol{o})}(\boldsymbol{o}, \boldsymbol{B})\big), \tag{15}$$

*where $g^*$ denotes the global minimum value of $g(\boldsymbol{o}, \boldsymbol{B})$, $g^{*(\boldsymbol{B})}(\boldsymbol{o}, \boldsymbol{B})$ is the minimum of $g(\boldsymbol{o}, \boldsymbol{B})$ with $\boldsymbol{B}$ fixed, and $f^{*(\boldsymbol{o})}(\boldsymbol{o}, \boldsymbol{B})$ is the minimum of $f(\boldsymbol{o}, \boldsymbol{B})$ with $\boldsymbol{o}$ fixed.*

**Assumption 3.** *For arbitrary $\boldsymbol{o}$ and $\boldsymbol{B}$, we have $|f(\boldsymbol{o}, \boldsymbol{B}) - g(\boldsymbol{o}, \boldsymbol{B})| \leq \sigma$.*

**Remark 1.** *Assumptions 1 and 2 are frequently invoked in the theoretical analysis of gradient-based optimization methods (Malinovsky et al., 2024; Zhou, 2018; Khaled & Richtárik, 2020), while Assumption 3 ensures that the smoothed loss function does not deviate excessively from the original function. This requirement is reasonable, as studies on the approximation of step functions have already demonstrated that the error between a smoothed function and a step function can be made small (Kyurkchiev & Markov, 2015; Iliev et al., 2015).*

## 5.2 CONVERGENCE RESULTS

Based on the above assumptions, we conduct a convergence analysis of Algorithm 1 and examine whether the stability of the algorithm is significantly affected by the surrogate gradient.

**Theorem 2.** *Suppose Assumptions 1–3 hold, and let $0 < \eta_1 \leq \min\big(\frac{1}{L_1}, \frac{1}{L_1\mu_1}\big)$, $0 < \eta_2 \leq \min\big(\frac{1}{L_2}, \frac{1}{L_2\mu_3}\big)$, $\boldsymbol{o}_k^{(0)} = \boldsymbol{o}_{k-1}$, $\boldsymbol{o}_k^{(N_1)} = \boldsymbol{o}_k$, $\boldsymbol{B}_k^{(0)} = \boldsymbol{B}_{k-1}$, and $\boldsymbol{B}_k^{(N_2)} = \boldsymbol{B}_k$, then for Algorithm 1, the following inequality holds:*

$$f(\boldsymbol{o}_{k+1}, \boldsymbol{B}_{k+1}) - f^* \leq (1 - \mu_3\eta_2)(1 - \mu_1\eta_1)\big[f(\boldsymbol{o}_k, \boldsymbol{B}_k) - f^*\big] + C, \tag{16}$$

*where*

$$C = 2\sigma(1 - \mu_1\eta_1)(1 - \mu_3\eta_2) + 2\sigma + \frac{\eta_1}{2}(1 - \mu_3\eta_2)\|\nabla_{\boldsymbol{B}} g(\boldsymbol{o}_k, \boldsymbol{B}_k)\|^2$$
$$+ \frac{\mu_3\eta_2}{2\mu_2}\|\nabla_{\boldsymbol{o}} g^{*(\boldsymbol{o})}(\boldsymbol{o}_{k+1}, \boldsymbol{B})\|^2. \tag{17}$$

**Remark 2.** *Theorem 2 comprises a linear convergence term $(1-\mu_3\eta_2)(1-\mu_1\eta_1)\left[f(\boldsymbol{o}_k, \boldsymbol{B}_k) - f^*\right]$ and an error term $C$, resembling the results obtained in analyses such as (Nguyen et al., 2017; Yuan et al., 2024). We note that the error term is largely determined by the smoothed objective function. A suitable smoothing technique can ensure that the error constant $C$ remains sufficiently small, thereby enabling the iterative process of the algorithm to approach the global optimum. The detailed proof of Theorem 2 can be found in Appendix A.*

Table 1: Results on LLaMA-2-7B. "Time Cost" denotes the training time required for the method to obtain the SNN. "Grain" denotes the number of granularities.

| PPL Perf. ↓ | $T$ | Time Cost | Wikitext2 | C4 | Redpajama | Pile | Avg. PPL |
|---|---|---|---|---|---|---|---|
| LLaMA-2-7B | N/A | N/A | 5.47 | 6.97 | 5.61 | 4.63 | 5.67 |
| SpikeLLM | 8 | 5h 54m | 5.86 | 7.51 | 6.08 | 4.97 | 6.10 |
| TTFSFormer | 128 | N/A | 11.88 | 16.47 | 13.18 | 9.32 | 12.71 |
| LAS | | N/A | 34.26 | 40.39 | 32.18 | 20.10 | 31.73 |
| SpikedAttention | 8 | 2m 02s | 19.02 | 25.05 | 20.77 | 14.80 | 19.91 |
| Ours (Grain=2) | | 2m 01s | 6.71 | 8.96 | 7.23 | 5.74 | 7.16 |
| Ours (Grain=3) | | 2m 04s | 7.10 | 9.71 | 7.80 | 6.09 | 7.68 |
| LAS | | N/A | 6.05 | 7.88 | 6.37 | 5.13 | 6.36 |
| SpikedAttention | 10 | 2m 28s | 11.64 | 15.47 | 12.84 | 9.47 | 12.36 |
| Ours (Grain=2) | | 2m 25s | 5.50 | 7.05 | 5.68 | 4.67 | 5.73 |
| Ours (Grain=3) | | 2m 27s | 5.53 | 7.06 | 5.69 | 4.68 | 5.74 |
| **ACC Perf. ↑** | $T$ | **Time Cost** | **WinoGrande** | **ArcC** | **ArcE** | **PiQA** | **Avg. ACC** |
| LLaMA-2-7B | N/A | N/A | 69.06 | 46.33 | 74.54 | 79.05 | 67.25 |
| SpikeLLM | 8 | 5h 54m | 67.40 | 42.58 | 71.46 | 77.75 | 64.80 |
| TTFSFormer | 128 | N/A | 70.56 | 44.88 | 73.11 | 78.94 | 66.87 |
| LAS | | N/A | 69.46 | 45.56 | 73.65 | 77.97 | 66.66 |
| SpikedAttention | 8 | 2m 02s | 68.19 | 41.38 | 68.77 | 77.20 | 63.89 |
| Ours (Grain=2) | | 2m 01s | 70.56 | 46.16 | 73.99 | 77.97 | 67.17 |
| Ours (Grain=3) | | 2m 04s | 70.96 | 46.08 | 74.33 | 77.86 | 67.31 |
| LAS | | N/A | 70.64 | 45.90 | 73.95 | 78.24 | 67.18 |
| SpikedAttention | 10 | 2m 28s | 67.32 | 40.10 | 68.73 | 76.44 | 63.15 |
| Ours (Grain=2) | | 2m 25s | 70.48 | 46.50 | 73.91 | 78.29 | 67.30 |
| Ours (Grain=3) | | 2m 27s | 70.88 | 46.16 | 73.78 | 78.13 | 67.24 |

# 6 EXPERIMENTS

## 6.1 EXPERIMENTAL SETUP

**Baseline.** We consider the following three baselines: **Full-precision ANN**, evaluated under the zero-shot setting, provides a standard performance reference for spiking LLMs. **SpikeLLM** (Xing et al., 2025) integrates SNNs with quantized ANNs to build a spike-driven large language model. **TTFSFormer** (Zhao et al., 2025) applies time-to-first-spike coding to transformer architecture and achieves the spiking transformer based on temporal coding. **LAS** (Chen et al., 2025b) employs SNN neurons with $\theta(t) = h(t) = d(t) = \tau \cdot 2^{-t}$ for ANN-to-SNN conversion, which can be viewed as a special instance of our method without optimization. **SpikedAttention** (Hwang et al., 2024) leverages single-spike phase coding to construct a spiking transformer, and we extend this design to the LLaMA model in this work. The implementation details of all methods are provided in Appendix D.1.

**Datasets and Metrics.** To effectively evaluate different methods, we adopt perplexity and accuracy as evaluation metrics. For perplexity, we conduct evaluations on Wikitext2 (Merity et al., 2016), C4 (Raffel et al., 2020), RedPajama (Weber et al., 2024), and Pile (Gao et al., 2020). For accuracy, we evaluate zero-shot reasoning performance on WinoGrande (Sakaguchi et al., 2021), ArcC, ArcE (Clark et al., 2018), and PiQA (Bisk et al., 2020). We report the accuracy (acc) for WinoGrande

and the accuracy norm for ArcC, ArcE, and PiQA. All accuracies are measured using lm_eval v0.4.2 (Sutawika et al., 2024).

## 6.2 MAIN RESULTS

We report the results of our method under different granularities, along with comparisons to other baselines. As shown in Table 1 and 2, our method delivers optimal overall performance by maintaining accuracy close to that of ANNs alongside low perplexity, highlighting the superiority of our paradigm over existing baselines. Notably, compared to LAS, which represents a special case of our method without optimization, our method yields a significant perplexity reduction, validating the effectiveness of the proposed neuron training algorithm. With respect to SpikedAttention, even with relaxation of its single-spike phase coding (as described in Appendix D.1), it fails to sustain satisfactory performance in our limited total timestep experimental setting ($T \in \{6, 8, 10\}$). Relative to SpikeLLM, our method demonstrates a substantial advantage in time cost, as its training process entails forward and backward propagation through the decoder layers, whereas our paradigm entirely eliminates this overhead. For TTFSFormer, our method can achieve comparable performance with fewer timesteps due to its time-to-first-spike coding. Furthermore, due to the enhanced representational capacity of multi-granularity phase coding, our method achieves ANN-to-SNN conversion for LLMs with a limited total timestep. The ablation study and the results on a larger-scale LLM are provided in the Appendix D.2 and D.3, respectively.

Table 2: Results on LLaMA-3-8B. "Time Cost" denotes the training time required for the method to obtain the SNN. "Grain" denotes the number of granularities.

| PPL Perf. ↓ | $T$ | Time Cost | Wikitext2 | C4 | Redpajama | Pile | Avg. PPL |
|---|---|---|---|---|---|---|---|
| LLaMA-3-8B | N/A | N/A | 6.14 | 8.88 | 7.44 | 5.52 | 7.00 |
| SpikeLLM | 8 | 6h 13m | >100 | >100 | >100 | >100 | >100 |
| TTFSFormer | 128 | N/A | 6.72 | 9.82 | 8.12 | 6.03 | 7.67 |
| LAS | | N/A | 93.13 | >100 | >100 | >100 | >100 |
| SpikedAttention | 6 | 1m 38s | >100 | >100 | >100 | 83.95 | >100 |
| Ours (Grain=2) | | 1m 36s | 8.04 | 12.25 | 10.14 | 7.23 | 9.42 |
| Ours (Grain=3) | | 1m 35s | 8.53 | 13.18 | 10.85 | 7.73 | 10.07 |
| LAS | | N/A | 7.06 | 10.49 | 8.57 | 6.51 | 8.16 |
| SpikedAttention | 8 | 2m 02s | 9.46 | 14.02 | 12.08 | 8.17 | 10.93 |
| Ours (Grain=2) | | 2m 01s | 6.32 | 9.14 | 7.69 | 5.74 | 7.22 |
| Ours (Grain=3) | | 1m 58s | 6.37 | 9.22 | 7.73 | 5.79 | 7.28 |
| **ACC Perf. ↑** | $T$ | Time Cost | WinoGrande | ArcC | ArcE | PiQA | Avg. ACC |
| LLaMA-3-8B | N/A | N/A | 72.85 | 53.33 | 77.74 | 80.85 | 71.19 |
| SpikeLLM | 8 | 6h 13m | 69.38 | 49.23 | 73.11 | 78.67 | 67.60 |
| TTFSFormer | 128 | N/A | 72.69 | 52.90 | 77.65 | 79.33 | 70.64 |
| LAS | | N/A | 71.19 | 51.88 | 73.48 | 79.33 | 68.97 |
| SpikedAttention | 6 | 1m 38s | 63.30 | 32.25 | 54.00 | 66.21 | 53.94 |
| Ours (Grain=2) | | 1m 36s | 73.16 | 47.87 | 73.74 | 77.64 | 68.10 |
| Ours (Grain=3) | | 1m 35s | 73.24 | 49.23 | 73.82 | 76.82 | 68.28 |
| LAS | | N/A | 74.82 | 54.52 | 77.48 | 80.74 | 71.89 |
| SpikedAttention | 8 | 2m 02s | 69.53 | 49.91 | 74.33 | 77.20 | 67.74 |
| Ours (Grain=2) | | 2m 01s | 72.69 | 54.35 | 78.11 | 80.14 | 71.32 |
| Ours (Grain=3) | | 1m 58s | 73.09 | 54.10 | 78.41 | 80.30 | 71.48 |

## 6.3 ENERGY ANALYSIS

To estimate energy consumption, we adopt the following theoretical energy estimation approach according to Rathi & Roy (2020); Li et al. (2021); Zhou et al. (2022); Deng et al. (2024); Zhao et al. (2025); Wang et al. (2022); Chen et al. (2025a;b),

$$E_{total} = E_{MAC} \cdot Count_{MAC} + E_{AC} \cdot Count_{AC}, \tag{18}$$

where $E_{MAC}$ and $E_{AC}$ denote the energy consumption of a single MAC and AC operation, respectively, while $Count_{MAC}$ and $Count_{AC}$ denote the counted numbers of MAC and AC operations during model inference. We measure MAC and AC operations using $E_{MAC} \approx 4.6pJ$ and $E_{AC} \approx 0.9pJ$, as reported in 45 nm CMOS technology (Horowitz, 2014). The results indicate that, for a single sample, the ANN-based LLM consumes 18.00 $J$, whereas our distribution-aware spiking LLM consumes only 10.44$\sim$10.46 $J$ as summarized in Table 3 ,i.e., our spiking LLM achieves a 42.0% reduction in energy consumption of MAC and AC operations compared to its ANN counterpart. Furthermore, we anticipate that continued progress in neuromorphic hardware will further enhance the efficiency of our spiking LLM, potentially leading to even larger reductions in both operations and energy consumption.

Table 3: The calculation count and the energy cost of ANN and our spiking LLMs with $T = 6$.

| Model | Method | Avg. ACC | Avg. PPL | Calculation Count | Energy Cost ($J$) |
|---|---|---|---|---|---|
| | ANN | 71.19 | 7.00 | 3912.08G MACs + 0.17G ACs | 18.00 |
| LLaMA-3-8B | Ours (Grain=2) | 68.10 | 9.42 | 15.87G MACs + 11521.88G ACs | 10.44 |
| | Ours (Grain=3) | 68.28 | 10.07 | 15.87G MACs + 11539.14G ACs | 10.46 |

## 7 CONCLUSION

To overcome latent conversion error arising from distribution misalignment, we propose multi-granularity phase coding, enabling SNN neurons to allocate mapped discrete values adaptively with respect to the activation distribution. Building on this coding scheme, we introduce a novel ANN-to-SNN conversion paradigm that leverages a cost-efficient alternating optimization neuron training algorithm to minimize conversion errors with respect to activation distributions. In future research, we intend to further advance our ANN-to-SNN conversion paradigm based on multi-granularity phase coding, targeting a smaller total timestep and improved energy efficiency.

## ACKNOWLEDGMENTS

Dr. Zhaogeng Liu was supported by the Young Scientists Fund (C Class) of the National Natural Science Foundation of China under Grant No. 62506142.

## ETHICS STATEMENT

All participants in this work, as well as the paper submission, adhere to the ICLR Code of Ethics ( https://iclr.cc/public/CodeOfEthics).

## REPRODUCIBILITY STATEMENT

We affirm that the results of this work are fully reproducible. Appendix A provides the theoretical proofs. Appendix D.1 details the experimental implementations, and the source code is available at `https://github.com/njzhenghy/SpikingLLM`.

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

# APPENDIX

## A  THEORETICAL ANALYSIS

### A.1  DEFINITION OF NOTATION

Table 4: Notation

| Notation | Description |
|---|---|
| $f(\boldsymbol{h}, \boldsymbol{\theta}, \boldsymbol{B})$ | Objective function. |
| $g(\boldsymbol{h}, \boldsymbol{\theta}, \boldsymbol{B})$ | Smoothed objective function. |
| $\boldsymbol{o}_k$ or $\boldsymbol{B}_k$ | The parameter value at the $k$-th iteration. |
| $\boldsymbol{d}$ | Output weight parameters of SNN neurons. |
| $\boldsymbol{B}$ | The phase variable to be optimized. |
| $\boldsymbol{h}$ | Reset strength parameters of SNN neurons. |
| $\boldsymbol{\theta}$ | Threshold parameters of SNN neurons. |
| $\boldsymbol{o}$ | Treat $\boldsymbol{h}$ and $\boldsymbol{\theta}$ as a single parameter $\boldsymbol{o}$. |
| $g^{*(\boldsymbol{B})}(\boldsymbol{o}, \boldsymbol{B})$ | The minimum of $g(\boldsymbol{o}, \boldsymbol{B})$ with $\boldsymbol{B}$ fixed. |
| $f^{*(\boldsymbol{o})}(\boldsymbol{o}, \boldsymbol{B})$ | The minimum of $f(\boldsymbol{o}, \boldsymbol{B})$ with $\boldsymbol{o}$ fixed. |
| $\eta_1$ | Learning rate for updating $\boldsymbol{h}$ and $\boldsymbol{\theta}$ (for updating $\boldsymbol{o}$). |
| $\eta_2$ | Learning rate for updating $\boldsymbol{B}$. |
| $L_1$ | Lipschitz constant of function $g(\boldsymbol{o}, \boldsymbol{B})$ w.r.t. variable $\boldsymbol{o}$. |
| $L_2$ | Lipschitz constant of function $f(\boldsymbol{o}, \boldsymbol{B})$ w.r.t. variable $\boldsymbol{B}$. |
| $\mu$ | PL condition constant. |
| $\sigma$ | Maximum error between $f(\boldsymbol{o}, \boldsymbol{B})$ and $g(\boldsymbol{o}, \boldsymbol{B})$ under identical parameters. |

### A.2  PROOFS

**Lemma 1.** *If Assumption 3 holds, then $|f^* - g^*| \le \sigma$, where $f^*$ denotes the global minimum value of $f(\boldsymbol{o}, \boldsymbol{B})$.*

**Proof of Lemma 1**:

If $f^* = f(\boldsymbol{o}_i, \boldsymbol{B}_i)$ and Assumption 3 hold, we have,

$$g^* \le g(\boldsymbol{o}_i, \boldsymbol{B}_i) \le f(\boldsymbol{o}_i, \boldsymbol{B}_i) + \sigma = f^* + \sigma. \tag{19}$$

Similarly, if $g^* = g(\boldsymbol{o}_j, \boldsymbol{B}_j)$ we have,

$$f^* \le f(\boldsymbol{o}_j, \boldsymbol{B}_j) \le g(\boldsymbol{o}_j, \boldsymbol{B}_j) + \sigma = g^* + \sigma. \tag{20}$$

So we have,

$$|f^* - g^*| \le \sigma. \tag{21}$$

**Proof of Theorem 2**:

From Assumption 1, we have,

$$g(\boldsymbol{o}_{k+1}, \boldsymbol{B}_k) \le g(\boldsymbol{o}_k, \boldsymbol{B}_k) + \langle \nabla_{\boldsymbol{o}} g(\boldsymbol{o}_k, \boldsymbol{B}_k), \boldsymbol{o}_{k+1} - \boldsymbol{o}_k \rangle + \frac{L_1}{2} \|\boldsymbol{o}_{k+1} - \boldsymbol{o}_k\|_2^2. \tag{22}$$

Due to $\boldsymbol{o}_{k+1} = \boldsymbol{o}_k - \eta_1 \nabla_{\boldsymbol{o}} g(\boldsymbol{o}_k, \boldsymbol{B}_k)$, we obtain,

$$g(\boldsymbol{o}_{k+1}, \boldsymbol{B}_k) \le g(\boldsymbol{o}_k, \boldsymbol{B}_k) - \eta_1 \|\nabla_{\boldsymbol{o}} g(\boldsymbol{o}_k, \boldsymbol{B}_k)\|_2^2 + \frac{L_1 \eta_1^2}{2} \|\nabla_{\boldsymbol{o}} g(\boldsymbol{o}_k, \boldsymbol{B}_k)\|_2^2$$

$$= g(\boldsymbol{o}_k, \boldsymbol{B}_k) + (\frac{L_1 \eta_1^2}{2} - \eta_1) \|\nabla_{\boldsymbol{o}} g(\boldsymbol{o}_k, \boldsymbol{B}_k)\|_2^2. \tag{23}$$

Let $0 < \eta_1 \le \min(\frac{1}{L_1}, \frac{1}{L_1 \mu_1})$, we have,

$$g(\boldsymbol{o}_{k+1}, \boldsymbol{B}_k) \le g(\boldsymbol{o}_k, \boldsymbol{B}_k) - \frac{\eta_1}{2} \|\nabla_{\boldsymbol{o}} g(\boldsymbol{o}_k, \boldsymbol{B}_k)\|_2^2. \tag{24}$$

From Assumption 2, we have,

$$\|\nabla g(\boldsymbol{o}, \boldsymbol{B})\|^2 = \|\nabla_{\boldsymbol{o}} g(\boldsymbol{o}, \boldsymbol{B})\|^2 + \|\nabla_{\boldsymbol{B}} g(\boldsymbol{o}, \boldsymbol{B})\|^2 \geq 2\mu_1 [g(\boldsymbol{o}, \boldsymbol{B}) - g^*]. \tag{25}$$

Rearranging the above equation, we have,

$$\|\nabla_{\boldsymbol{o}} g(\boldsymbol{o}, \boldsymbol{B})\|^2 \geq 2\mu_1 [g(\boldsymbol{o}, \boldsymbol{B}) - g^*] - \|\nabla_{\boldsymbol{B}} g(\boldsymbol{o}, \boldsymbol{B})\|^2. \tag{26}$$

$$g(\boldsymbol{o}_{k+1}, \boldsymbol{B}_k) \leq g(\boldsymbol{o}_k, \boldsymbol{B}_k) - \mu_1 \eta_1 [g(\boldsymbol{o}_k, \boldsymbol{B}_k) - g^*] + \frac{\eta_1}{2} \|\nabla_{\boldsymbol{B}} g(\boldsymbol{o}_k, \boldsymbol{B}_k)\|^2. \tag{27}$$

Subtracting $g^*$ from both sides, we have,

$$\begin{aligned}
g(\boldsymbol{o}_{k+1}, \boldsymbol{B}_k) - g^* &\leq g(\boldsymbol{o}_k, \boldsymbol{B}_k) - g^* - \mu_1 \eta_1 [g(\boldsymbol{o}_k, \boldsymbol{B}_k) - g^*] + \frac{\eta_1}{2} \|\nabla_{\boldsymbol{B}} g(\boldsymbol{o}_k, \boldsymbol{B}_k)\|^2 \\
&= (1 - \mu_1 \eta_1) [g(\boldsymbol{o}_k, \boldsymbol{B}_k) - g^*] + \frac{\eta_1}{2} \|\nabla_{\boldsymbol{B}} g(\boldsymbol{o}_k, \boldsymbol{B}_k)\|^2.
\end{aligned} \tag{28}$$

Next, we prove the inequality result obtained when updating $\boldsymbol{B}$. From Assumption 1, we have,

$$f(\boldsymbol{o}_{k+1}, \boldsymbol{B}_{k+1}) \leq f(\boldsymbol{o}_{k+1}, \boldsymbol{B}_k) + \langle \nabla_{\boldsymbol{B}} f(\boldsymbol{o}_{k+1}, \boldsymbol{B}_k), \boldsymbol{B}_{k+1} - \boldsymbol{B}_k \rangle + \frac{L_2}{2} \|\boldsymbol{B}_{k+1} - \boldsymbol{B}_k\|_2^2. \tag{29}$$

Due to $\boldsymbol{B}_{k+1} = \boldsymbol{B}_k - \eta_2 \nabla_{\boldsymbol{B}} f(\boldsymbol{o}_{k+1}, \boldsymbol{B}_k)$, we obtain,

$$\begin{aligned}
f(\boldsymbol{o}_{k+1}, \boldsymbol{B}_{k+1}) &\leq f(\boldsymbol{o}_{k+1}, \boldsymbol{B}_k) - \eta_2 \|\nabla_{\boldsymbol{B}} f(\boldsymbol{o}_{k+1}, \boldsymbol{B}_k)\|_2^2 + \frac{L_2 \eta_2^2}{2} \|\nabla_{\boldsymbol{B}} f(\boldsymbol{o}_{k+1}, \boldsymbol{B}_k)\|_2^2 \\
&= f(\boldsymbol{o}_{k+1}, \boldsymbol{B}_k) + (\frac{L_2 \eta_2^2}{2} - \eta_2) \|\nabla_{\boldsymbol{B}} f(\boldsymbol{o}_{k+1}, \boldsymbol{B}_k)\|_2^2.
\end{aligned} \tag{30}$$

Let $0 < \eta_2 \leq \min(\frac{1}{L_2}, \frac{1}{L_2 \mu_3})$, we have,

$$f(\boldsymbol{o}_{k+1}, \boldsymbol{B}_{k+1}) \leq f(\boldsymbol{o}_{k+1}, \boldsymbol{B}_k) - \frac{\eta_2}{2} \|\nabla_{\boldsymbol{B}} f(\boldsymbol{o}_{k+1}, \boldsymbol{B}_k)\|_2^2. \tag{31}$$

From Assumption 2, we have,

$$\|\nabla_{\boldsymbol{B}} f(\boldsymbol{o}_{k+1}, \boldsymbol{B}_k)\|_2^2 \geq 2\mu_3 \left[ f(\boldsymbol{o}_{k+1}, \boldsymbol{B}_k) - f^{*(\boldsymbol{o})}(\boldsymbol{o}_{k+1}, \boldsymbol{B}) \right]. \tag{32}$$

Substituting Equation (32) into Equation (31), we obtain,

$$f(\boldsymbol{o}_{k+1}, \boldsymbol{B}_{k+1}) \leq f(\boldsymbol{o}_{k+1}, \boldsymbol{B}_k) - \mu_3 \eta_2 \left[ f(\boldsymbol{o}_{k+1}, \boldsymbol{B}_k) - f^{*(\boldsymbol{o})}(\boldsymbol{o}_{k+1}, \boldsymbol{B}) \right]. \tag{33}$$

Subtracting $f^*$ from both sides, we have,

$$\begin{aligned}
f(\boldsymbol{o}_{k+1}, \boldsymbol{B}_{k+1}) - f^* &\leq f(\boldsymbol{o}_{k+1}, \boldsymbol{B}_k) - f^* - \mu_3 \eta_2 \left[ f(\boldsymbol{o}_{k+1}, \boldsymbol{B}_k) - f^{*(\boldsymbol{o})}(\boldsymbol{o}_{k+1}, \boldsymbol{B}) \right] \\
&= (1 - \mu_3 \eta_2) [f(\boldsymbol{o}_{k+1}, \boldsymbol{B}_k) - f^*] + \mu_3 \eta_2 \left[ f^{*(\boldsymbol{o})}(\boldsymbol{o}_{k+1}, \boldsymbol{B}) - f^* \right].
\end{aligned} \tag{34}$$

Next, we combine the results obtained above. Applying Assumption 3 and Lemma 1 to Equation (28), we have,

$$f(\boldsymbol{o}_{k+1}, \boldsymbol{B}_k) - f^* - 2\sigma \leq (1 - \mu_1 \eta_1) [f(\boldsymbol{o}_k, \boldsymbol{B}_k) - f^* + 2\sigma] + \frac{\eta_1}{2} \|\nabla_{\boldsymbol{B}} g(\boldsymbol{o}_k, \boldsymbol{B}_k)\|^2, \tag{35}$$

$$\begin{aligned}
f(\boldsymbol{o}_{k+1}, \boldsymbol{B}_k) - f^* &\leq (1 - \mu_1 \eta_1) [f(\boldsymbol{o}_k, \boldsymbol{B}_k) - f^* + 2\sigma] + \frac{\eta_1}{2} \|\nabla_{\boldsymbol{B}} g(\boldsymbol{o}_k, \boldsymbol{B}_k)\|^2 + 2\sigma \\
&= (1 - \mu_1 \eta_1) [f(\boldsymbol{o}_k, \boldsymbol{B}_k) - f^*] + 2\sigma(2 - \mu_1 \eta_1) + \frac{\eta_1}{2} \|\nabla_{\boldsymbol{B}} g(\boldsymbol{o}_k, \boldsymbol{B}_k)\|^2.
\end{aligned} \tag{36}$$

Substituting Equation (36) into Equation (34), we obtain,

$$\begin{aligned}
f(\boldsymbol{o}_{k+1}, \boldsymbol{B}_{k+1}) - f^* &\leq (1 - \mu_3 \eta_2)(1 - \mu_1 \eta_1) [f(\boldsymbol{o}_k, \boldsymbol{B}_k) - f^*] + 2\sigma(2 - \mu_1 \eta_1)(1 - \mu_3 \eta_2) \\
&\quad + \frac{\eta_1}{2}(1 - \mu_3 \eta_2) \|\nabla_{\boldsymbol{B}} g(\boldsymbol{o}_k, \boldsymbol{B}_k)\|^2 + \mu_3 \eta_2 \left[ f^{*(\boldsymbol{o})}(\boldsymbol{o}_{k+1}, \boldsymbol{B}) - f^* \right].
\end{aligned} \tag{37}$$

Since $f^{*(o)}(\boldsymbol{o}_{k+1}, \boldsymbol{B}) - f^* \leq g^{*(o)}(\boldsymbol{o}_{k+1}, \boldsymbol{B}) - g^* + 2\sigma$ and from Assumption 2, we have,

$$f^{*(o)}(\boldsymbol{o}_{k+1}, \boldsymbol{B}) - f^* \leq \frac{\|\nabla_o g^{*(o)}(\boldsymbol{o}_{k+1}, \boldsymbol{B})\|^2}{2\mu_2} + 2\sigma. \tag{38}$$

Substituting Equation (38) into Equation (36), we obtain,

$$f(\boldsymbol{o}_{k+1}, \boldsymbol{B}_{k+1}) - f^* \leq (1 - \mu_3\eta_2)(1 - \mu_1\eta_1)\left[f(\boldsymbol{o}_k, \boldsymbol{B}_k) - f^*\right] + 2\sigma(2 - \mu_1\eta_1)(1 - \mu_3\eta_2)$$
$$+ \frac{\eta_1}{2}(1 - \mu_3\eta_2)\|\nabla_{\boldsymbol{B}} g(\boldsymbol{o}_k, \boldsymbol{B}_k)\|^2 + \frac{\mu_3\eta_2\|\nabla_o g^{*(o)}(\boldsymbol{o}_{k+1}, \boldsymbol{B})\|^2}{2\mu_2} + 2\sigma\mu_3\eta_2. \tag{39}$$

Simplifying yields,

$$f(\boldsymbol{o}_{k+1}, \boldsymbol{B}_{k+1}) - f^* \leq (1 - \mu_3\eta_2)(1 - \mu_1\eta_1)\left[f(\boldsymbol{o}_k, \boldsymbol{B}_k) - f^*\right] + C, \tag{40}$$

where

$$C = 2\sigma(1 - \mu_1\eta_1)(1 - \mu_3\eta_2) + 2\sigma + \frac{\eta_1}{2}(1 - \mu_3\eta_2)\|\nabla_{\boldsymbol{B}} g(\boldsymbol{o}_k, \boldsymbol{B}_k)\|^2$$
$$+ \frac{\mu_3\eta_2}{2\mu_2}\|\nabla_o g^{*(o)}(\boldsymbol{o}_{k+1}, \boldsymbol{B})\|^2. \tag{41}$$

The above corresponds to the case where both the inner iteration counts of $\boldsymbol{o}$ and $\boldsymbol{B}$ are equal to one. We now consider the case where the inner iteration count of $\boldsymbol{o}$ is $N_1$, and that of $\boldsymbol{B}$ is $N_2$. In fact, the above result still holds when the number of inner iterations is not equal to one. This is because, based on Assumptions 1 and 2, and by employing an argument similar to that used in deriving Equation (33), we can establish the following inequality:

$$g(\boldsymbol{o}_{k+1}^{(j_1+1)}, \boldsymbol{B}_k^{(N_2)}) - g^{*(\boldsymbol{B})}(\boldsymbol{o}, \boldsymbol{B}_k^{(N_2)}) \leq (1 - \mu_1\eta_1)\left[g(\boldsymbol{o}_{k+1}^{(j_1)}, \boldsymbol{B}_k^{(N_2)}) - g^{*(\boldsymbol{B})}(\boldsymbol{o}, \boldsymbol{B}_k^{(N_2)})\right], \tag{42}$$

$$f(\boldsymbol{o}_{k+1}^{(N_1)}, \boldsymbol{B}_{k+1}^{(j_2+1)}) - f^{*(o)}(\boldsymbol{o}_{k+1}^{(N_1)}, \boldsymbol{B}) \leq (1 - \mu_3\eta_2)\left[f(\boldsymbol{o}_{k+1}^{(N_1)}, \boldsymbol{B}_{k+1}^{(j_2)}) - f^{*(o)}(\boldsymbol{o}_{k+1}^{(N_1)}, \boldsymbol{B})\right]. \tag{43}$$

Since $0 < \eta_1 \leq \min(\frac{1}{L_1}, \frac{1}{L_1\mu_1})$ and $0 < \eta_2 \leq \min(\frac{1}{L_2}, \frac{1}{L_2\mu_3})$, let $\boldsymbol{o}_k^{(0)} = \boldsymbol{o}_{k-1}$, $\boldsymbol{o}_k^{(N_1)} = \boldsymbol{o}_k$, $\boldsymbol{B}_k^{(0)} = \boldsymbol{B}_{k-1}$ and $\boldsymbol{B}_k^{(N_2)} = \boldsymbol{B}_k$ we have,

$$g(\boldsymbol{o}_{k+1}^{(j_1+1)}, \boldsymbol{B}_k) < g(\boldsymbol{o}_{k+1}^{(j_1)}, \boldsymbol{B}_k), \tag{44}$$

$$f(\boldsymbol{o}_{k+1}, \boldsymbol{B}_{k+1}^{(j_2+1)}) < f(\boldsymbol{o}_{k+1}, \boldsymbol{B}_{k+1}^{(j_2)}). \tag{45}$$

By applying the above equation, we obtain,

$$g(\boldsymbol{o}_{k+1}, \boldsymbol{B}_k) = g(\boldsymbol{o}_{k+1}^{(N_1)}, \boldsymbol{B}_k) < g(\boldsymbol{o}_{k+1}^{(1)}, \boldsymbol{B}_k), \tag{46}$$

$$f(\boldsymbol{o}_{k+1}, \boldsymbol{B}_{k+1}) = f(\boldsymbol{o}_{k+1}, \boldsymbol{B}_{k+1}^{(N_2)}) < f(\boldsymbol{o}_{k+1}, \boldsymbol{B}_{k+1}^{(1)}). \tag{47}$$

For the case in which the number of inner iterations differs from one, applying Equations (46) and (47) yields a result analogous to Equations (34) and (36), with the distinction that $\boldsymbol{o}_k$ and $\boldsymbol{B}_k$ here represent the values at the end of each inner loop. We have,

$$f(\boldsymbol{o}_{k+1}, \boldsymbol{B}_{k+1}) - f^* \leq f(\boldsymbol{o}_{k+1}, \boldsymbol{B}_{k+1}^{(1)}) - f^*$$
$$\leq (1 - \mu_3\eta_2)\left[f(\boldsymbol{o}_{k+1}, \boldsymbol{B}_k) - f^*\right] + \mu_3\eta_2\left[f^{*(o)}(\boldsymbol{o}_{k+1}, \boldsymbol{B}) - f^*\right], \tag{48}$$

$$f(\boldsymbol{o}_{k+1}, \boldsymbol{B}_k) - f^* \leq f(\boldsymbol{o}_{k+1}^{(1)}, \boldsymbol{B}_k) - f^*$$
$$\leq (1 - \mu_1\eta_1)\left[f(\boldsymbol{o}_k, \boldsymbol{B}_k) - f^*\right] + 2\sigma(2 - \mu_1\eta_1) + \frac{\eta_1}{2}\|\nabla_{\boldsymbol{B}} g(\boldsymbol{o}_k, \boldsymbol{B}_k)\|^2. \tag{49}$$

Based on Equations (48) and (49), we obtain exactly the same result as in Equation (39).

## B  CONVERT NONLINEAR OPERATION IN LLM

**Attention Layer.** The attention architecture of our method is presented as follows:

$$\mathbf{Q} \approx \sum_{t=1}^{T} \mathbf{Q}_{s,t}, \ \mathbf{K} \approx \sum_{t=1}^{T} \mathbf{K}_{s,t}, \ \mathbf{V} \approx \sum_{t=1}^{T} \mathbf{V}_{s,t}, \tag{50}$$

where $\mathbf{Q}$, $\mathbf{K}$ and $\mathbf{V}$ denote the query $\mathbf{Q}$, key $\mathbf{K}$ and value $\mathbf{V}$, and $\mathbf{Q}_{s,t}$, $\mathbf{K}_{s,t}$ and $\mathbf{V}_{s,t}$ denote the spiking query, key and value at timestep $t$.

Then, we need to enable the Activation-Activation (AA) multiplication in attention within SNN, which occurs between the query $\mathbf{Q}$ and key $\mathbf{K}$ as well as attention array $\mathbf{A} = \mathbf{QK}$ and value $\mathbf{V}$. Fortunately, You et al. (2024) have paved the way for such AA multiplication. Specifically, taking the multiplication between query and key as an example, it can be written as:

$$\mathbf{A} = \mathbf{Q} \cdot \mathbf{K} \approx \sum_{t=1}^{T} \mathbf{Q}_{s,t} \cdot \sum_{t=1}^{T} \mathbf{K}_{s,t}$$

$$= \sum_{t=1}^{T} \left( \mathbf{S}_{Q,t} \cdot \mathbf{K}_{s,t} + \mathbf{Q}_{s,t} \cdot \mathbf{S}_{K,t} - \mathbf{Q}_{s,t} \cdot \mathbf{K}_{s,t} \right), \tag{51}$$

where $\mathbf{S}_{Q,t}$ and $\mathbf{S}_{K,t}$ represent the accumulated spike output of query and key from $1$ to $t$. Therefore, the result of AA multiplication at each time $t$ is $\mathbf{S}_{Q,t} \cdot \mathbf{K}_{s,t} + \mathbf{Q}_{s,t} \cdot \mathbf{S}_{K,t} - \mathbf{Q}_{s,t} \cdot \mathbf{K}_{s,t}$.

**Spiking Softmax, Spiking RMSNorm and Spiking SiLU Activation.** Inspired by the literature (You et al., 2024), we use the following process to enable Softmax, RMSnorm, and SiLU activation in SNN.

$$\mathbf{I}(t) = \mathbf{I}(t-1) + I(t), \tag{52}$$
$$\mathbf{O}(t) = \phi(\mathbf{I}(t)), \tag{53}$$
$$O(t) = \mathbf{O}(t) - \mathbf{O}(t-1), \tag{54}$$

where $\mathbf{I}(t)$ is the accumulated input at $t$ timestep; $I(t)$ is the input at $t$ timestep; $\phi(\cdot)$ is the Softmax, RMSnorm and SiLU activation and $O(t)$ is the output at $t$ timestep.

**Spiking MLP.** Except for the linear layers in MLP, the most important operation is the Activation-Activation Hadamard product, which exists between the output of a linear layer and the output of the spiking SiLU function. It can be written as

$$\mathbf{A} \odot \mathbf{B} = \sum_{t=1}^{T} \mathbf{A}_t \odot \sum_{t=1}^{T} \mathbf{B}_t \tag{55}$$

$$= \sum_{t=1}^{T} \left( \mathbf{A}_t \odot \mathbf{B}_t + \sum_{i=1,i\neq t}^{T} \frac{\mathbf{A}_t \odot \mathbf{B}_i + \mathbf{A}_i \odot \mathbf{B}_t}{2} \right),$$

where $\mathbf{A}$ denotes the output of the **up_proj** and $\mathbf{B}$ denotes the output of Spiking SiLU. Therefore, the result of Hadamard product at each time $t$ is $\mathbf{A}_t \odot \mathbf{B}_t + \sum_{i=1,i\neq t}^{T} \frac{\mathbf{A}_t \odot \mathbf{B}_i + \mathbf{A}_i \odot \mathbf{B}_t}{2}$.

## C  ADAPTIVE GRANULARITY ALLOCATION

We propose a differentiable adaptive search algorithm for granularity allocation in phase encoding. In particular, we select the optimal granularity allocation based on the data distribution. To achieve this, we perform 10,000 down-sampling operations on the activation values during the training process of LLaMA-2 and LLaMA-3 in the experiment. Furthermore, we leverage the differentiable search framework presented in (Liu et al., 2018) to compute the Softmax of the different granularity allocation weights, as follows:

$$\overline{o}(x) = \sum_{o \in \mathcal{O}} \frac{\exp(\alpha_o)}{\sum_{o' \in \mathcal{O}} \exp(\alpha_{o'})} o(x). \tag{56}$$

Here, $\mathcal{O}$ represents the set of granularity candidates, and $\alpha_o$ represents the granularity allocation coefficients. Based on this, our optimization objective can be formulated as:

$$\min_{\boldsymbol{\alpha}} \ \mathcal{L}_{MSE}(\boldsymbol{\alpha}; \boldsymbol{w}), \tag{57}$$

$$\text{s.t.} \quad \boldsymbol{w}^* = \arg\min_{\boldsymbol{w}} \ \mathcal{L}_{MSE}(\boldsymbol{w}; \boldsymbol{\alpha}), \tag{58}$$

where $\boldsymbol{w}(\boldsymbol{h}, \boldsymbol{\theta}; \boldsymbol{B})$ represents the parameters of the SNN neurons. We obtain the optimal architecture parameter $\boldsymbol{\alpha}^*$ as shown in Algorithm 2, and subsequently retrain the SNN neuron with the optimal granularity allocation.

---

**Algorithm 2** Differentiable Architecture Search with Adaptive Granularity Allocation

---

1: **Input:** Training dataset $\boldsymbol{X}$, neuron patameter $\boldsymbol{w} = (\boldsymbol{h}, \boldsymbol{\theta}; \boldsymbol{B})$, learning rate $\eta_t$, optimization steps $N_\alpha$.
2: **for** $i = 1, \cdots, N_\alpha$ **do**
3:     Obtain $\boldsymbol{w}^*$ by using Alogrithem 1 under $N_1 = N_2 = 1$.
4:     Update $\boldsymbol{\alpha}$ by $\boldsymbol{\alpha} = \boldsymbol{\alpha} - \nabla_{\boldsymbol{\alpha}}\mathcal{L}_{MSE}(\boldsymbol{\alpha}; \boldsymbol{w}^*, \boldsymbol{X})$.
5: **end for**
6: **return** $\boldsymbol{\alpha}^*$.

---

## D  MORE EXPERIMENTAL RESULTS

### D.1  IMPLEMENTATION DETAILS

We conduct experiments on a server equipped with multiple 80GB NVIDIA A100 GPUs. For the full-precision ANNs of the LLaMA family, we use open-source models from the HuggingFace and evaluate their performance under the FP16 setting. For SpikeLLM, we employ the released open-source implementation and assess its performance in the W4A8 configuration (4-bit weight and 8-bit activation). For TTFSFormer, since the original implementation is not available on LLaMA, we implemented a simplified version by applying time-to-first-spike coding within our code framework. For LAS, since its available open-source implementation is not adapted to the LLaMA model, we construct a simplified implementation by setting the SNN neuron parameters as $\theta(t) = h(t) = d(t) = \tau \cdot 2^{-t}$ and regard it as the special case of our method without optimization. For the same reason, we also implement a simplified version of SpikedAttention in our code framework by setting the SNN neuron parameters as $\theta(t) = h(t) = d(t) = \tau \cdot 2^{-t}$ and applying the single-spike technique (Hwang & Kung, 2024). It is worth noting that under a limited total timestep experimental setting, the single-spike technique leads to a collapse in performance. Therefore, we relax this technique to allow two spikes instead of one. To handle outliers in LLMs, we apply the Hadamard rotation and prefixed outlier tokens techniques introduced in the literature (Chen et al., 2024).

### D.2  ABLATION STUDY

**Effectiveness of Multi-Granularity.** To verify the effectiveness of multi-granularity in our proposed phase coding, we perform an ablation study varying the number of granularities, and the results are presented in Table 5. We observe that the best accuracy and perplexity are not achieved with a single granularity, which demonstrates the effectiveness of our design. It is worth noting that increasing the granularity does not necessarily lead to better results. When we increase the number of granularities, the model can non-uniformly allocate discrete values more flexibly. The solution space with Grain = 2 or 3 strictly contains the Grain = 1 solution space, so, in principle, more granularities can only help. However, in practice, this larger solution space also makes optimization more non-convex and prone to local minima. Our ablation studies also confirm exactly this trade-off. When we push granularity to the extreme (e.g., setting the number of granularities equal to the timestep $T$), the performance actually is not the best, indicating that excessive granularity makes the optimization harder and the solution is more likely to be suboptimal.

**Interaction between the Timestep $T$ and the Number of Granularities.** We observe that the effect of granularity depends on the timestep $T$, and we should study this more systematically. Specifically, we provide bases $\boldsymbol{B}$ and training loss curves for different combinations of timestep and

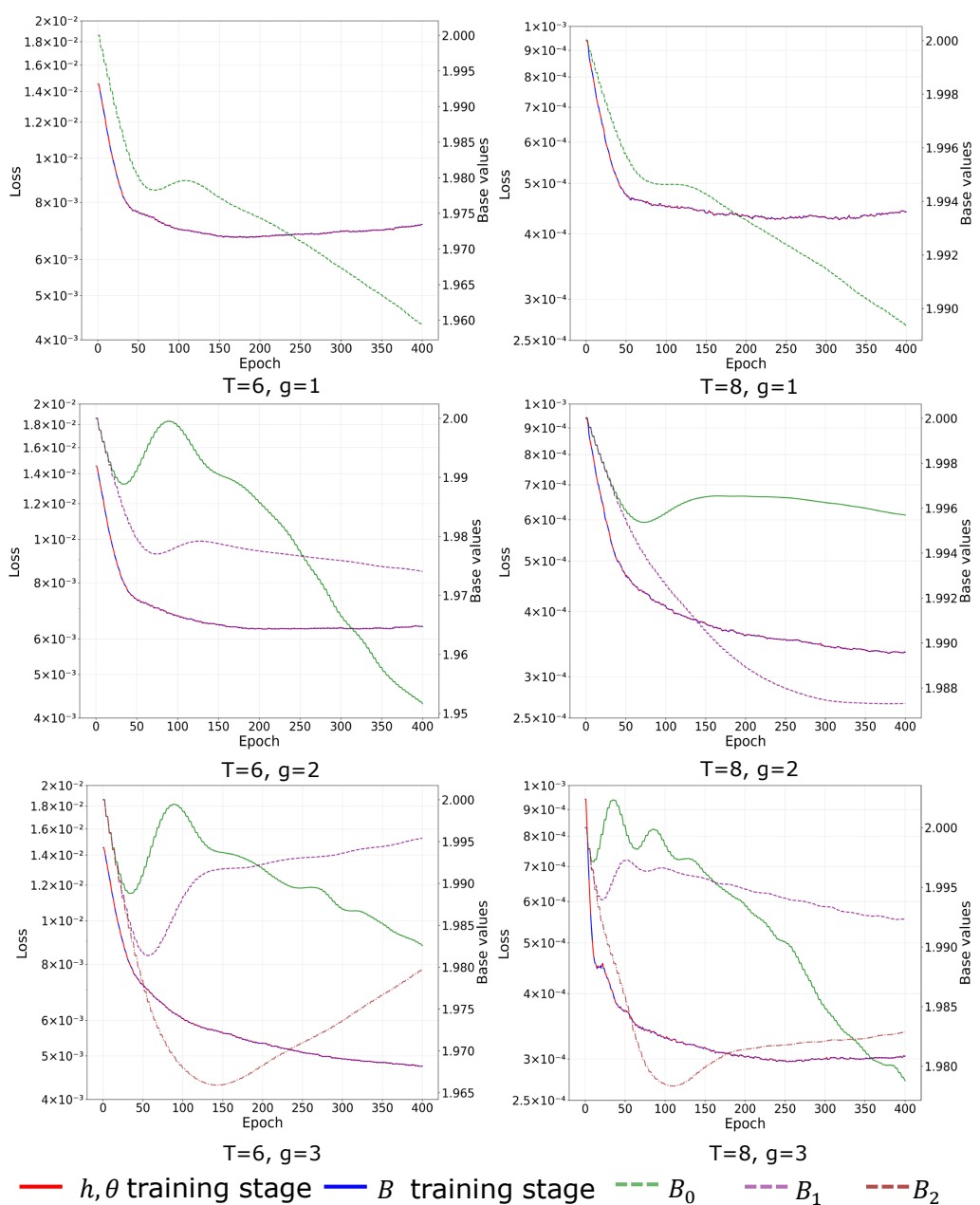

Figure 4: The alternating red and blue curves trace the loss dynamics during staged optimization, with red intervals indicating updates to $h$ and $\theta$, and blue intervals corresponding to $B$ training. Dashed curves denote the evolution of base values across granularities $B_0$, $B_1$, and $B_2$.

granularity, showing how the optimization converges. These curves in Figure 4 clearly illustrate the evolution of $\boldsymbol{B}$ and the training loss during the optimization process under different timesteps and numbers of granularities. Conceptually, in our phase-coding neuron, the number of representable discrete values grows as $2^T$. When $T$ is large, the discrete representation is already quite dense, so redistributing these discrete values via multi-granularity provides smaller gains. This explains why, in Tables 1, 2, and 9, the improvement from increasing Grain at a larger T appears modest. In contrast, when $T$ is lower, the total number of discrete values is more limited, so where these values are placed becomes much more critical. In this regime, multi-granularity can reduce conversion error by allocating more resolution to high-density regions of the activation distribution.

Table 5: Ablation study on the number of granularities with $T = 8$.

| Models | Grain | Avg. ACC | Avg. PPL |
|--------|-------|----------|----------|
| LLaMA-2-7B | 1 | 67.03 | 7.01 |
| | 2 | 67.17 | 7.16 |
| | 3 | 67.31 | 7.68 |
| | $T$ | 67.14 | 7.44 |
| LLaMA-3-8B | 1 | 71.09 | 7.24 |
| | 2 | 71.32 | 7.22 |
| | 3 | 71.48 | 7.28 |
| | $T$ | 71.40 | 7.27 |

**Comparison between Joint Optimization and Alternating Optimization.** To demonstrate the effectiveness of alternating optimization, we additionally conduct an ablation study using joint optimization. In this setting, we observed a worse downstream performance in Table 6. We believe this is because, without decoupling the two stages, updates to $B$ and $h, \theta$ interfere with each other. In our method, these parameters play different roles. The bases $B$ determine the distribution of discrete representable values. By using multiple bases, we shape how discrete values are distributed to better match the activation distribution. The neuron parameters $h, \theta$ determine how a given continuous input is mapped to one of those discrete values, i.e., how the spike dynamics choose which discrete value is used. We explicitly separate them so that our method to effectively minimize the conversion error arising from distribution misalignment. Moreover, from the perspective of the convergence of alternating optimization, the loss function is non-differentiable with respect to $h$ and $\theta$, so backpropagation for these variables must rely on surrogate gradients, whereas the optimization of $B$ can directly use the true gradients. Therefore, intuitively, if $h$, $\theta$, and $B$ are optimized simultaneously, the errors introduced by the surrogate gradients will propagate to the updates of $B$, thereby amplifying the overall optimization error and reducing the stability of convergence.

Table 6: Results on joint optimization and alternating optimization. "PPL" denotes the perplexity on Wikitext2.

| Model | $T$/Grain | Method | WinoGrande | ArcC | ArcE | PiQA | PPL |
|-------|-----------|--------|------------|------|------|------|-----|
| LLaMA-2-7B | 8/2 | Joint | 70.09 | 45.22 | 73.99 | 77.86 | 6.58 |
| | | Alter | 70.56 | 46.16 | 73.99 | 77.97 | 6.71 |
| | 8/3 | Joint | 69.85 | 45.65 | 73.78 | 77.69 | 6.41 |
| | | Alter | 70.96 | 46.08 | 74.33 | 77.86 | 7.10 |
| LLaMA-3-8B | 6/2 | Joint | 72.38 | 47.18 | 71.38 | 76.55 | 7.61 |
| | | Alter | 73.16 | 47.87 | 73.74 | 77.64 | 8.04 |
| | 6/3 | Joint | 71.67 | 47.18 | 72.39 | 74.54 | 7.52 |
| | | Alter | 73.24 | 49.23 | 73.82 | 76.82 | 8.53 |

**Decoupling $h$ and $\theta$ from Each Other.** In conventional formulations, $h$ and $\theta$ are often tied (e.g., $h = \theta$), which reduces the degrees of freedom of the neuron dynamics. In our setting, once the discrete values (determined by $B$) are fixed, the neuron still needs enough flexibility to shape the mapping from continuous activations to these values. By allowing $h$ and $\theta$ to vary independently, we can increase the expressive power of the neuron dynamics and enable a finer adjustment of the mapping between continuous activations and discrete values. Empirically, as shown in Table 7, we observe that this extra flexibility helps reduce the approximation error between the SNN neuron output and the original ANN activation.

**Weight Quantization.** We apply weight quantization to the LLaMA-2-7B and recompute the activation distributions under 8-bit and 4-bit weights. As shown in Figure 5, we observe that while quantization slightly changes the exact shape of the distributions, the activations remain highly non-uniform and layer-dependent, so the core motivation of our distribution-aware design still holds. We also evaluate our distribution-aware multi-granularity phase coding under quantized weights, including 8-bit and 4-bit settings. The results in Table 8 show that our approach maintains competitive

Table 7: Results on whether to decouple $h$ and $\theta$. "PPL" denotes the perplexity on Wikitext2.

| Model | $T$/Grain | Decouple | WinoGrande | ArcC | ArcE | PiQA | PPL |
|---|---|---|---|---|---|---|---|
| LLaMA-2-7B | 8/2 | No | 70.24 | 45.65 | 74.03 | 77.86 | 7.62 |
| | | Yes | 70.56 | 46.16 | 73.99 | 77.97 | 6.71 |
| | 8/3 | No | 70.17 | 45.90 | 73.86 | 77.97 | 6.90 |
| | | Yes | 70.96 | 46.08 | 74.33 | 77.86 | 7.10 |
| LLaMA-3-8B | 6/2 | No | 71.98 | 45.90 | 71.80 | 75.03 | 8.00 |
| | | Yes | 73.16 | 47.87 | 73.74 | 77.64 | 8.04 |
| | 6/3 | No | 73.40 | 47.78 | 72.26 | 75.35 | 7.63 |
| | | Yes | 73.24 | 49.23 | 73.82 | 76.82 | 8.53 |

performance under 8-bit and even 4-bit weights, demonstrating that our method is compatible with weight quantization.

Table 8: Results on LLaMA-2-7B with weight quantization. "PPL" denotes the perplexity on Wikitext2.

| Method | $T$ | Weight Bit | WinoGrande | ArcC | ArcE | PiQA | PPL |
|---|---|---|---|---|---|---|---|
| LLaMA-2-7B | N/A | 16 | 69.06 | 46.33 | 74.54 | 79.05 | 5.47 |
| Ours (Grain=2) | 8 | 8 | 70.09 | 45.48 | 73.82 | 77.64 | 7.56 |
| Ours (Grain=3) | | | 70.40 | 45.82 | 74.16 | 77.53 | 7.25 |
| Ours (Grain=2) | 8 | 4 | 67.80 | 42.75 | 71.21 | 76.88 | 8.91 |
| Ours (Grain=3) | | | 68.51 | 43.26 | 71.04 | 77.15 | 8.41 |

Table 9: Results on LLaMA-2-13B. "Time Cost" denotes the training time required for the method to obtain the SNN. "Grain" denotes the number of granularities.

| PPL Perf. ↓ | $T$ | Time Cost | Wikitext2 | C4 | Redpajama | Pile | Avg. PPL |
|---|---|---|---|---|---|---|---|
| LLaMA-2-13B | N/A | N/A | 4.88 | 6.47 | 5.19 | 4.34 | 5.22 |
| SpikeLLM | 8 | 10h 41m | 5.20 | 6.91 | 5.57 | 4.63 | 5.58 |
| LAS | 8 | N/A | 18.02 | 21.82 | 17.02 | 11.85 | 17.18 |
| SpikedAttention | | 2m 35s | 8.90 | 12.97 | 10.74 | 8.52 | 10.28 |
| Ours (Grain=2) | | 2m 35s | 5.07 | 6.74 | 5.39 | 4.51 | 5.43 |
| Ours (Grain=3) | | 2m 34s | 5.29 | 7.40 | 5.91 | 4.83 | 5.86 |
| LAS | 10 | N/A | 5.03 | 6.76 | 5.38 | 4.48 | 5.41 |
| SpikedAttention | | 3m 05s | 6.43 | 8.54 | 6.94 | 5.80 | 6.93 |
| Ours (Grain=2) | | 3m 05s | 4.90 | 6.54 | 5.23 | 4.38 | 5.26 |
| Ours (Grain=3) | | 3m 06s | 4.90 | 6.54 | 5.23 | 4.37 | 5.26 |
| **ACC Perf. ↑** | $T$ | Time Cost | WinoGrande | ArcC | ArcE | PiQA | Avg. ACC |
| LLaMA-2-13B | N/A | N/A | 72.45 | 49.15 | 77.44 | 80.52 | 69.89 |
| SpikeLLM | 8 | 10h 41m | 69.30 | 47.27 | 76.22 | 79.05 | 67.96 |
| LAS | 8 | N/A | 72.77 | 51.28 | 77.27 | 80.14 | 70.37 |
| SpikedAttention | | 2m 35s | 72.38 | 45.82 | 74.54 | 78.07 | 67.70 |
| Ours (Grain=2) | | 2m 35s | 73.24 | 50.43 | 77.31 | 80.47 | 70.36 |
| Ours (Grain=3) | | 2m 34s | 73.72 | 50.60 | 77.53 | 80.14 | 70.50 |
| LAS | 10 | N/A | 72.53 | 50.17 | 77.10 | 80.85 | 70.16 |
| SpikedAttention | | 3m 05s | 70.40 | 44.88 | 74.20 | 78.24 | 66.93 |
| Ours (Grain=2) | | 3m 05s | 72.69 | 49.91 | 77.06 | 81.01 | 70.17 |
| Ours (Grain=3) | | 3m 06s | 72.77 | 50.00 | 77.10 | 80.74 | 70.15 |

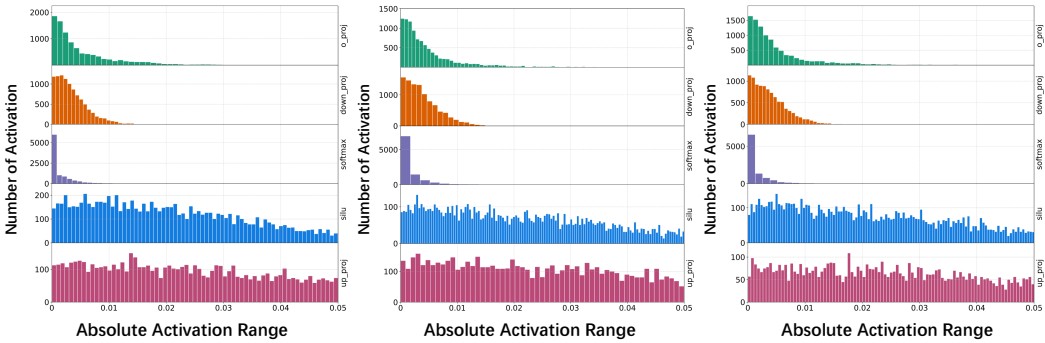

Figure 5: Activation distribution after weight quantization.

## D.3 RESULTS ON LARGER-SCALE LLM

In Table 9, we provide a comparison of our method with other baselines on a larger-scale LLM (LLaMA-2-13B). Similar phenomena are observed on LLaMA-2-13B as on LLaMA-2-7B and LLaMA-3-8B, which fully demonstrates that our method remains effective for larger-scale LLMs.

## D.4 RESULTS ON MULTIMODAL MODEL

Evaluating our method beyond language models can further strengthen the empirical evidence for its effectiveness. To this end, we have extended our distribution-aware multi-granularity phase coding from LLMs to a multimodal model, which is structurally and functionally different from language models. Specifically, we extend our method to CLIP and evaluate the performance of the spiking CLIP model on image classification tasks. The results in Table 10 show that our method can be successfully applied in this setting as well.

Table 10: Performance Comparison Results on ImageNet, CIFAR10, and CIFAR100 using CLIP model. "FP32" represents the performance of the ANN evaluated under the float32 precision.

| Model | Method | $T$ | ImageNet | CIFAR10 | CIFAR100 | Avg. ACC |
|---|---|---|---|---|---|---|
| ViT-B/32 | FP32 | N/A | 57.71 | 89.69 | 64.01 | 70.47 |
| | LAS | 8 | 55.42 | 89.27 | 66.22 | 70.30 |
| | Ours (Grain=2) | 8 | 56.72 | 90.48 | 66.11 | 71.10 |
| | Ours (Grain=3) | 8 | 56.78 | 90.23 | 65.64 | 70.88 |
| ViT-B/16 | FP32 | N/A | 63.42 | 90.82 | 67.07 | 73.77 |
| | LAS | 8 | 58.87 | 84.59 | 59.77 | 67.74 |
| | Ours (Grain=2) | 8 | 60.68 | 89.70 | 65.49 | 71.96 |
| | Ours (Grain=3) | 8 | 61.22 | 89.77 | 65.25 | 72.08 |
| ViT-L/14 | FP32 | N/A | 71.13 | 95.82 | 76.41 | 81.12 |
| | LAS | 8 | 69.71 | 88.82 | 70.29 | 76.27 |
| | Ours (Grain=2) | 8 | 69.61 | 94.99 | 77.61 | 80.74 |
| | Ours (Grain=3) | 8 | 69.63 | 94.87 | 77.17 | 80.56 |

## D.5 RESULTS ON OTHER LLM

To enhance the completeness of our method, we add additional experiments on Qwen2-7B (Team et al., 2024) using our proposed multi-granularity phase coding. The results in Table 11 show that our method maintains high performance on Qwen2-7B, further demonstrating its effectiveness and scalability.

Table 11: Results on Qwen2-7B. "Grain" denotes the number of granularities. "PPL" denotes the perplexity on Wikitext2.

| Method | T | WinoGrande | ArcC | ArcE | PiQA | Avg. Acc | PPL |
|---|---|---|---|---|---|---|---|
| Qwen2-7B | N/A | 72.38 | 49.91 | 74.71 | 81.23 | 69.56 | 7.14 |
| LAS | 8 | 70.96 | 50.60 | 74.20 | 80.52 | 69.07 | 10.18 |
| SpikedAttention | 8 | 61.96 | 28.33 | 48.96 | 65.13 | 51.10 | >100 |
| Ours (Grain=2) | 8 | 73.40 | 50.60 | 74.12 | 81.01 | 69.78 | 7.41 |
| Ours (Grain=3) | 8 | 72.53 | 50.60 | 74.07 | 80.85 | 69.51 | 7.42 |

## D.6 MORE RESULTS AT A LOWER TIMESTEP

To further demonstrate that our method is scalable to a lower timestep, we also include experiments with T=6 on Llama-2-7B, and the results are in Table 12. Our method significantly reduces perplexity compared to all baselines without sacrificing accuracy.

Table 12: Results on LLaMA-2-7B with $T = 6$. "Grain" denotes the number of granularities. "PPL" denotes the perplexity on Wikitext2.

| Method ↑ | T | WinoGrande | ArcC | ArcE | PiQA | Avg. Acc | PPL |
|---|---|---|---|---|---|---|---|
| LLaMA-2-7B | N/A | 69.06 | 46.33 | 74.54 | 79.05 | 67.25 | 5.47 |
| LAS | | 67.96 | 44.28 | 72.52 | 77.86 | 65.65 | 45.50 |
| SpikedAttention | 6 | 66.69 | 41.64 | 70.03 | 76.77 | 63.78 | 50.05 |
| Ours (Grain=2) | | 67.64 | 45.31 | 72.26 | 77.58 | 65.70 | 12.19 |
| Ours (Grain=3) | | 68.98 | 44.37 | 72.52 | 77.86 | 65.93 | 10.79 |

## D.7 MORE ENERGY ANALYSIS

For the energy comparison with other Spiking LLM, we report the energy consumption data for SpikeLLM on LLaMA-3-8B. For the energy consumption calculations of both SpikeLLM and our Spiking LLMs, we employ identical configurations and perform a statistical analysis of the MACs and ACs generated by the same components. The results in Table 13 demonstrate that our method achieves lower energy consumption compared to SpikeLLM.

Table 13: The calculation count and the energy cost of ANN, SpikeLLM, and our method on LLaMA-3-8B.

| Method | Calculation Count | Energy Cost ($J$) |
|---|---|---|
| ANN | 3912.08G MACs + 0.17G ACs | 18.00 |
| SpikeLLM | 2.79G MACs + 14507.31G ACs | 13.87 |
| Ours (Grain=2) | 15.87G MACs + 11521.88G ACs | 10.44 |
| Ours (Grain=3) | 15.87G MACs + 11539.14G ACs | 10.46 |

The memory access and data movement are the primary sources of energy consumption (which we refer to as the read/write cost) on existing hardware (Dampfhoffer et al., 2022). In order to further validate the effectiveness of our method, we expand our energy analysis to explicitly include the costs associated with read/write. Specifically, for the calculation of energy consumed by reading and writing weights and activations, we refer to the energy estimation approach for both ANN and SNN models presented in Hwang et al. (2024). We set the 32-bit read/write energy for weights and activations, $E_{read}$ and $E_{write}$, to 5 $pJ$ and conducted a comparison of the energy consumption between ANN and our spiking LLM. The total energy equations for ANN and SNN are given by:

$$E_{total}^{ANN} = 2E_{read} \cdot Count_{read} + E_{write} \cdot Count_{write} + E_{MAC} \cdot Count_{MAC} + E_{AC} \cdot Count_{AC}$$

$$E_{total}^{SNN} = E_{neuron} + (1 + 1/32)E_{read} \cdot \sum_t Count_{read}^t + 1/32 \cdot E_{write} \cdot \sum_t Count_{write}^t$$
$$+ E_{MAC} \cdot Count_{MAC} + E_{AC} \cdot Count_{AC}$$

$$(59)$$

Where $E_{neuron}$ represents the energy consumption of a neuron, and $E_{read}$ and $E_{write}$ denote the energy consumption for read and write operations, respectively. $Count_{read}$ and $Count_{write}$ denote the number of read and write operations. The factors of 1/32 in the SNN formula are due to the fact that activations in SNNs are represented using the 1-bit spike. Table 14 and Table 17 show that memory access accounts for at least 68% of the total energy consumption in both ANN and current SNN models, making it the primary source of energy consumption. Nevertheless, thanks to the sparsity inherent in SNN computations, our results show that the total energy consumption of our method remains over 12% lower than that of the ANN.

Table 14: Energy consumption of LLaMA-3-8B under ANN and our method, including both read and write operations.

| Method | Read/Write Cost ($J$) | MAC & AC Cost ($J$) | Total Energy Cost ($J$) | Total Energy Cost Relative to ANN |
|--------|----------------------|---------------------|------------------------|-----------------------------------|
| ANN | 38.47 | 18.00 | 56.47 | 100.00% |
| Ours (Grain=2) | 38.35 | 10.44 | 48.84 | 86.48% |
| Ours (Grain=3) | 38.93 | 10.46 | 49.43 | 87.53% |

Table 15: Relative energy consumption of LLaMA-3-8B under ANN and our spiking LLMs, including both read and write operations.

| Method | Proportion of Read/Write Cost | Proportion of MAC & AC Cost |
|--------|-------------------------------|------------------------------|
| ANN | 68.12% | 31.88% |
| Ours (Grain=2) | 78.52% | 21.38% |
| Ours (Grain=3) | 78.76% | 21.16% |

As supported by Dampfhoffer et al. (2022), the energy consumption of SNNs is closely tied to the spike firing rate. To address this issue, we propose a masking mechanism. Specifically, we exploit the characteristic of phase coding, where the encoding value decreases as the timestep increases. Consequently, spikes from neurons that fire early can be considered redundant, and those occurring at later timesteps can be discarded. This strategy effectively reduces the spike firing rate by eliminating redundant spikes without significantly impacting performance. As a result, the increased activation sparsity leads to a substantial reduction in the overall energy consumption of the SNN. In our energy estimation, we also include the cost of the masking operation. To be cautious, we upper-bound this cost by assigning the mask the same energy as a full neuron-level computation. Nevertheless, even under this assumption, the mask-related cost still accounts for only a small fraction of the total SNN energy, as neuron computation contributes relatively little compared with data movement and memory access. The lower spike firing rate resulting from the masking operation ultimately yields a lower value for $\tilde{Count}$ than for $Count$ in the energy calculation. The results of energy consumption with mask are in Table 16 and Table 17.

$$E_{total}^{SNN} = 2 \cdot E_{neuron} + (1 + 1/32)E_{read} \cdot \sum_t \tilde{Count}_{read}^t + 1/32 \cdot E_{write} \cdot \sum_t \tilde{Count}_{write}^t$$
$$+ E_{MAC} \cdot \tilde{Count}_{MAC} + E_{AC} \cdot \tilde{Count}_{AC}$$

$$(60)$$

Where $\tilde{Count}_{read}$ and $\tilde{Count}_{write}$ represent the number of read and write operations, and $\tilde{Count}_{MAC}$ and $\tilde{Count}_{AC}$ represent the number of MAC and AC operations, all after reducing the spike firing rate.

Table 16: Energy Consumption of LLaMA-3-8B for ANN and our spiking LLMs with mask, including both read and write operations. "PPL" denotes the perplexity on Wikitext2.

| Method | Avg. ACC | PPL | Read/Write Cost ($J$) | MAC & AC Cost ($J$) | Total Energy Cost ($J$) | Total Energy Cost Relative to ANN |
|---|---|---|---|---|---|---|
| ANN | 71.19 | 6.14 | 38.47 | 18.00 | 56.47 | 100.00% |
| Ours (Grain=2) | 66.18 | 8.82 | 34.02 | 6.08 | 40.13 | 71.03% |
| Ours (Grain=3) | 66.50 | 9.55 | 34.35 | 6.14 | 40.53 | 71.77% |

Table 17: Relative energy consumption of LLaMA-3-8B under ANN and our spiking LLMs with mask, including both read and write operations.

| Method | Proportion of Read/Write Cost | Proportion of MAC & AC Cost |
|---|---|---|
| ANN | 68.12% | 31.88% |
| Ours (Grain=2) | 84.77% | 15.15% |
| Ours (Grain=3) | 84.75% | 15.15% |

# E    USE OF LLMS

In this work, LLMs are employed solely for polishing or grammar checking text that is originally written by us.

