# OpenReview forum: "Distribution-Aware Multi-Granularity Phase Coding: Towards Lower Conversion Error for Spike-Driven Large Language Models"
_ICLR.cc/2026/Conference — ICLR 2026 Poster_

### Official Review · Reviewer_wbNY · 2025-10-26

**Soundness:** 2
**Presentation:** 3
**Contribution:** 2
**Rating:** 4
**Confidence:** 4

**Summary:**

This paper addresses the potential conversion error caused by activation distribution mismatch when converting pre-trained ANN LLMs to SNN LLMs. The authors propose a distribution-aware multi-granularity phase coding approach, which aims to achieve a reasonable discrete value allocation by minimizing conversion error relative to the activation distributions.

**Strengths:**

1. Distribution-aware multi-granularity phase coding is a novel and well-motivated method (derived from quantization distortion in information theory), providing a new approach to solve the distribution mismatch problem in ANN-SNN conversion.

2. Experimental results (on LLaMA-2 and LLaMA-3 models) show that this method achieves SOTA conversion performance (close to the ANN's PPL and accuracy), and the time cost for conversion (i.e., training the neurons) is extremely low (within minutes).

**Weaknesses:**

I like the distribution-aware conversion ideas and appreciate the authors' effort in fine-tuning SNNs on LLMs within minutes, but my main concern lies in the energy claims：

 1. End-to-end energy saving claim: In both the abstract and Section 6.3, the authors declare that "On commodity GPUs," their SNNs achieve an "end-to-end" reduction compared with their ANN counterparts, which is a very strong claim. However, upon checking Table 3, I found that the energy saving calculation relies only on the count of MACs and ACs. The energy for memory access, data movement, and control logic is not considered. SNNs may suffer more from these overheads due to their long spike train representations and multiple weight accesses across the time window.
2.  Furthermore, on GPUs, QNNs benefit from regular memory access patterns that enable efficient data reuse opportunities, whereas SNNs suffer from sparse and irregular memory access patterns. Can the authors explain how their SNNs running on GPUs solve these problems?Thus, if the authors want to claim the "end-to-end" "42%" energy consumption reduction, I recommend that they either: Consider these hardware overheads and discuss how to implement SNNs efficiently on GPUs; Measure the actual power when running the SNN and ANN on their GPU and multiply it by the running time to show the actual energy consumption on GPUs; Run their models on neuromorphic chips like Loihi and compare the energy.
3. Also, the spiking Softmax and RMSNorm (Appendix B) seem to still have complex multiplication or exponentiation operations. Is there any way to do a simulation of these functions and also apply it to the ANN-to-SNN conversion paradigm? If not, on GPUs, was the energy consumption of these exp and sqrt operations, which run at every timestep T, included in the total energy calculation in Table 3?

Minor Points:

1. In Figure 1a, the meanings of the x and y axes should be labeled on the figure to be more intuitive.

2. In Figure 2, the dimensions of Q, K, and V should include the time domain (e.g., n*d*T).

3. For the energy comparison, the paper should also include a comparison with other SpikingLLM papers, like those mentioned in Table 1.

4. If the authors tried weight quantization, does the distribution of each layer change? What are the results for, e.g., 4-bit or 8-bit weights for Spiking LLMs, since the baseline SpikeLLM can reduce the weight to 4-bits?

5. Potential complexity of multi-granularity tuning: Although multi-granularity phase coding improves representational capacity, learning and optimizing multiple bases may be more complex than the traditional method with a single base. Can author discuss more about the energy overhead?

**Questions:**

See weakness above.

---

> ### Author Response · Authors · 2025-11-21
> **Response to Reviewer wbNY - Part 1**
>
> Thank you to the reviewers for the energy and time dedicated to our paper. You can find the responses to all weaknesses and questions below. We have also made revisions to the original paper, with the modified parts highlighted in blue.
>
> **W1 & W2 - part1: (i) SNNs may suffer more from these overheads due to their long spike train representations and multiple weight accesses across the time window.** **(ii) Consider these hardware overheads and discuss how to implement SNNs efficiently on GPUs; Measure the actual power when running the SNN and ANN on their GPU and multiply it by the running time to show the actual energy consumption on GPUs.**
>
> We thank the reviewer for raising this important point regarding memory access and hardware. We would like to address the concerns and clarify our framework.
>
>
> * **Clarification of Energy Consumption Terminology and Acknowledgment of Misleading Statements**
>
>   We acknowledge that the descriptions and conclusions in our paper regarding energy consumption are somewhat misleading (for example, we should not have used the term "end-to-end energy consumption"). We have corrected these misleading descriptions in the revised version. The revised sentences are in Section 6.3.
>
> * **Context of Energy Consumption in Brain-Inspired Hardware**
>
>   Thank the reviewer for correctly pointing out that modern GPUs vary greatly in energy-consumption design, and that memory access and data movement are key components that we did not explicitly account for in our original analysis. Existing LLM are implicitly designed for typical von Neumann architectures, where computation and storage are separated without considering the compute-in-memory characteristics of non-von Neumann neuromorphic chips [1,2].
>
>   By contrast, **our framework is intended as an architecture design for neuromorphic (non-von Neumann) chips.** Such brain-inspired hardware adopts compute-in-memory and event-driven execution, so the data representation and operator execution style are fundamentally different from those of conventional LLMs on GPUs. In this setting, MAC/AC operations are the primary contributors to energy consumption, while memory access and data movement are no longer the dominant terms. Therefore, we focus on counting MAC/AC operations and multiplying them by standard per-operation energy coefficients.
>
>   We fully agree that this MAC/AC-based estimate does not correspond to the true end-to-end energy of contemporary GPUs. We acknowledge the reviewer's valid concern, and it is important to note that we do not intend to downplay the significance of memory or data movement in general GPUs. However, for the context of neuromorphic SNN execution, MAC/AC operations are far more representative of the energy cost, and hence we chose to focus on them for the energy comparison.
>
> * **Mainstream Approach of MAC/AC Energy Consumption**
>
>   Regarding the approach used to estimate energy consumption for MAC/AC operations, we want to emphasize that this is the mainstream approach in energy estimation for SNN, as widely accepted in the field, and the energy-per-operation for MAC/AC is typically derived from standards based on 45 nm technology [3-10]. While this does not capture every aspect of energy consumption, it provides a reasonable approximation.
>
> We hope these clarifications help to address the reviewer's concerns. Thank you again for your valuable feedback.
>
> [1] Muir, Dylan Richard, and Sadique Sheik. "The road to commercial success for neuromorphic technologies." Nature communications 16.1 (2025): 3586.
>
> [2] Wan, Weier, et al. "A compute-in-memory chip based on resistive random-access memory." Nature 608.7923 (2022): 504-512.
>
> [3] Rathi, Nitin, and Kaushik Roy. "Diet-snn: Direct input encoding with leakage and threshold optimization in deep spiking neural networks." arXiv preprint arXiv:2008.03658 (2020).
>
> [4] Li, Yuhang, et al. "Differentiable spike: Rethinking gradient-descent for training spiking neural networks." Advances in neural information processing systems 34 (2021): 23426-23439.
>
> [5] Zhou, Zhaokun, et al. "Spikformer: When spiking neural network meets transformer." arXiv preprint arXiv:2209.15425 (2022).
>
> [6] Deng, Shikuang, et al. "Spiking Token Mixer: An event-driven friendly Former structure for spiking neural networks." Advances in Neural Information Processing Systems 37 (2024): 128825-128846.
>
> [7] Zhao, Lusen, et al. "TTFSFormer: A TTFS-based Lossless Conversion of Spiking Transformer." Forty-second International Conference on Machine Learning.
>
> [8] Wang, Yuchen, et al. "Signed Neuron with Memory: Towards Simple, Accurate and High-Efficient ANN-SNN Conversion." IJCAI. 2022.
>
> [9] Chen, Long, et al. "Fas: Fast ann-snn conversion for spiking large language models." arXiv preprint arXiv:2502.04405 (2025).
>
> [10] Chen, Long, Xiaotian Song, and Yanan Sun. "LAS: Loss-less ANN-SNN Conversion for Fully Spike-Driven Large Language Models." arXiv preprint arXiv:2505.09659 (2025).

---

> ### Author Response · Authors · 2025-11-21
> **Response to Reviewer wbNY - Part 2**
>
> **W1 & W2 - part2: Furthermore, on GPUs, QNNs benefit from regular memory access patterns that enable efficient data reuse opportunities, whereas SNNs suffer from sparse and irregular memory access patterns. Can the authors explain how their SNNs running on GPUs solve these problems?**
>
> * Thank you for raising this point. We fully agree that, given current GPU architectures, QNNs naturally align with the hardware: their regular memory-access patterns and dense computation map efficiently onto GEMM-optimized GPUs. In contrast, SNNs rely on sparse, event-driven activations that introduce irregular memory access, preventing their theoretical efficiency advantages from being fully realized on today’s GPU platforms, which are fundamentally designed for dense workloads.
>
> * In addition, the mainstream SNN frameworks [1,2] were never developed with the goal of replacing or competing with QNNs on GPUs, and therefore do not explicitly address challenges caused by sparse and irregular memory access—nor were they intended to. Our intent is similar: we do not argue that SNNs should outperform QNNs on GPUs in terms of raw throughput.
>
> * Along the same lines, this work aims to design a spiking LLM framework specifically optimized for neuromorphic chips, paving the way for future deployment and utilization in neuromorphic computing systems.
>
>
> [1] Fang, Wei, et al. "Spikingjelly: An open-source machine learning infrastructure platform for spike-based intelligence." Science Advances 9.40 (2023): eadi1480.
>
> [2] https://snntorch.readthedocs.io/en/latest/
>
> **W1 & W2 - part3: Run their models on neuromorphic chips like Loihi and compare the energy.**
> As for running our models on neuromorphic chips like Loihi, we fully agree that running spiking LLMs on real neuromorphic hardware and measuring actual energy consumption would be an ideal way to validate our framework. However, current generations of neuromorphic chips (e.g., Loihi, TrueNorth) are primarily designed for conventional, relatively small-scale neural networks, rather than transformer-based LLMs. In practice, both the available neuron/synapse budget and the software toolchains for these chips are not yet well-suited to support LLaMA-scale Transformer architectures. This makes it very difficult at the moment to faithfully deploy and evaluate our full spiking LLM models on such hardware. So we are unable to include these experiments at this stage. Instead, we follow the commonly used operation-based energy estimation (MAC/AC counts with standard per-operation costs) as an approximation.

---

> ### Author Response · Authors · 2025-11-21
> **Response to Reviewer wbNY - Part 3**
>
> **W3: Also, the spiking Softmax and RMSNorm (Appendix B) seem to still have complex multiplication or exponentiation operations. Is there any way to do a simulation of these functions and also apply it to the ANN-to-SNN conversion paradigm? If not, on GPUs, was the energy consumption of these exp and sqrt operations, which run at every timestep T, included in the total energy calculation in Table 3?**
>
> We thank the reviewer for raising this important point regarding the complex operations (multiplication and exponentiation) involved in spiking Softmax and RMSNorm.
>
> * We acknowledge that, in our current framework, the spiking Softmax and RMSNorm still involve multiplication and exponentiation (e.g., exp, sqrt), and we do not simulate these functions with purely spike-based, multiplication-free approximations. However, we would like to clarify their relative computational weight in the context of LLMs.
>
>   We observe that FLOPs and energy cost are primarily dominated by the matrix multiplications in the attention mechanism and MLP modules. From the perspective of MAC and AC computations, the computational cost of nonlinear functions such as Softmax and normalization layers typically accounts for only a small fraction of the total computation.
>   This makes the computational cost of Softmax and RMSNorm negligible compared to the cost of linear projections and attention/MLP matrix multiplications. Therefore, keeping these functions in their original form is acceptable and does not materially affect our conclusions about energy reduction based on MAC and AC operations.
>
>   Moreover, implementing these components in an ANN-to-SNN manner is compatible with our conversion framework. We fully agree that developing spike-based approximations for nonlinear functions such as Softmax and RMSNorm would further improve our spiking LLMs. Systematically simulating or approximating these functions within the SNN domain is an important and complementary direction, and we consider it a promising future work of the current paper, whose primary focus is on handling the core high-cost components of LLMs.
>
> * In our original calculation for Table 3, the dominant source of MAC/AC counts comes from the large matrix multiplications, and we only do not explicitly include the cost of the exp and sqrt operations in Softmax. In response to the reviewer's concern, we have now re-estimated their contribution. Following Reference [1], we approximate one exp as equivalent to 20 MACs, and one sqrt as equivalent to 12 MACs. Using these conversions, we compute the total operation-based energy. For the ANN, including the exp/sqrt operations of all nonlinear functions increases the estimated energy by only about 0.4%. For the SNN, the increase is below 2%. Both increments are very small compared to the overall energy budget and do not materially affect our main conclusion, i.e., the estimated energy reduction of about 42% remains essentially unchanged.
>
> Table: Proportion of Total MAC Operations for Nonlinear Operations in the Llama3 Model
> | Operation Type | MAC Proportion |
> |----------------|----------------|
> | RMSNorm         | 0.052%            |
> | SiLU         | 0.185%            |
> | Softmax         | 0.205%            |
>
>
>
> [1] Chen, Long, Xiaotian Song, and Yanan Sun. "LAS: Loss-less ANN-SNN Conversion for Fully Spike-Driven Large Language Models." arXiv preprint arXiv:2505.09659 (2025).
>
> **W4: In Figure 1a, the meanings of the x and y axes should be labeled on the figure to be more intuitive.**
>
> We thank the reviewer for this helpful suggestion. We agree that explicitly labeling the x and y axis meanings in Figure 1(a) will make the figure more intuitive and easier to understand. In the revised version of the paper, we have updated Figure 1(a) to clearly include the axis labels.
>
> **W5: In Figure 2, the dimensions of Q, K, and V should include the time domain (e.g., ndT).**
>
> We thank the reviewer for this careful observation. We agree that explicitly including the time dimension in the shapes of Q, K, and V will make the spiking attention pipeline clearer. In the revised version, we have updated Figure 2 to explicitly reflect the time domain in the dimensions of Q, K, and V.

---

> ### Author Response · Authors · 2025-11-21
> **Response to Reviewer wbNY - Part 4**
>
> **W6: For the energy comparison, the paper should also include a comparison with other SpikingLLM papers, like those mentioned in Table 1.**
>
> We thank the reviewer for this helpful suggestion. We agree that providing energy comparisons against other SpikingLLM methods can further strengthen the empirical evaluation of our approach.
>
> * Regarding LAS and SpikedAttention, all these methods (including ours) are implemented within the same codebase, and they share very similar overall computational pipelines for the LLaMA backbone. Therefore, the estimated energy consumption differences between these methods are expected to be relatively small. For this reason, we chose to focus our additional energy analysis on the baseline SpikeLLM.
>
> * ~~Due to the limited rebuttal period, we will update the energy comparison against SpikeLLM in the coming days.~~
>
> * We have updated the energy consumption data for SpikeLLM on LLaMA-3-8B. For the energy consumption calculations of both SpikeLLM and our spiking LLMs, we employed identical configurations and performed a statistical analysis of the MACs and ACs generated by the same components. The results demonstrate that our methods achieve lower energy consumption compared to SpikeLLM.
>
> Table: The calculation count and the energy cost of ANN, SpikeLLM and our spiking LLMs.
> |Method|Calculation Count| Energy Cost (J)|
> |:-:|:-:|:-:|
> |ANN|3912.08G MACs+0.17G ACs|18.00|
> |SpikeLLM|2.785G MACs+14507.31G ACs|13.87|
> |Ours(Grain=2)|15.87G MACs+11521.88G ACs|10.44|
> |Ours(Grain=3)|15.87G MACs+11539.14G ACs|10.46|
>
>
> **W7: If the authors tried weight quantization, does the distribution of each layer change? What are the results for, e.g., 4-bit or 8-bit weights for Spiking LLMs, since the baseline SpikeLLM can reduce the weight to 4-bits?**
>
> We thank the reviewer for this insightful question. We agree that understanding the interaction between weight quantization and activation distributions is important for Spiking LLMs.
>
> * In response, we have applied weight quantization to the LLM and re-computed the activation distributions under 8-bit and 4-bit weights. **The updated activation histograms after weight quantization have been added to the revised version** (Subsection Weight Quantization of Appendix D.2). We observe that while quantization slightly changes the exact shape of the distributions, the activations remain highly non-uniform and layer-dependent, so the core motivation of our distribution-aware design still holds.
>
> * We have also evaluated our distribution-aware multi-granularity phase coding under quantized weights (including 8-bit and 4-bit settings). The corresponding performance (PPL and zero-shot accuracy) for our framework with quantized weights has been reported in the revised manuscript. The results show that our approach **maintains competitive performance under 8-bit and even 4-bit weights**, demonstrating that our framework is compatible with weight quantization.
>
> Table: Results on LLaMA-2-7B with weight quantization. "PPL" denotes the perplexity on Wikitext2.
> | Method|T| Weight Bit| WinoGrande|ArcC| ArcE| PiQA|PPL|
> |-|-|-|-|-:|-:|-:|-:|
> | LLaMA-2-7B |N/A|16| 69.06|46.33|74.54|79.05| 5.47|
> |Ours (Grain=2)|8|8|70.09|45.48|73.82|77.64|7.56|
> |Ours (Grain=3)|8|8|70.40|45.82|74.16|77.53|7.25|
> |Ours (Grain=2)|8|4|67.80|42.75|71.21|76.88|8.91|
> |Ours (Grain=3)|8|4|68.51|43.26|71.04|77.15|8.41|
>
> **W8: Potential complexity of multi-granularity tuning: Although multi-granularity phase coding improves representational capacity, learning and optimizing multiple bases may be more complex than the traditional method with a single base. Can author discuss more about the energy overhead?**
>
> We thank the reviewer for raising this important point about the potential complexity and energy overhead of learning multiple bases in multi-granularity phase coding.
>
> * First, in our alternating optimization scheme, **the number of parameters optimized at each step is actually very small**. Moving from a single-granularity to a multi-granularity design only introduces a few additional scalar bases. The optimization only touches the neuron-specific parameters ($\boldsymbol{\theta}$, $\boldsymbol{h}$) and a handful of bases, rather than the full model weights.
>
> * Second, **the optimization process is decoupled from the LLM's forward pass**. We pre-collect activations from the ANN model once, and then perform neuron optimization offline using these saved activations, without repeatedly running the full model. As a result, the tuning procedure is very fast and resource-efficient; in practice, it runs in minutes and is dominated neither by computation nor by memory overhead.
>
> * Finally, **once the neuron parameters are optimized, they are fixed during inference**. The spiking LLM then performs forward computation with deterministic neuron parameters, without any extra runtime adaptation. Therefore, there is no additional energy overhead during inference compared to a single-base phase coding scheme.

---

> > ### Comment · Reviewer_wbNY · 2025-11-25
> >
> > Thanks for your detailed response that solves most of my concerns. However, my main concerns about energy efficiency still exist. As the author states in the rebuttal, “In this setting, MAC/AC operations are the primary contributors to energy consumption”. Could you provide a reference, such as a pie chart or energy breakdown from similar works, to support this? Usually, when running a neuromorphic chip, MAC/AC energy accounts for less than 10-30% of total energy consumption and is not the primary contributor in most cases [1, 2, 3]. Furthermore, since most neuromorphic chips use low-bit precision, the computation energy is often very low compared to data movement or memory access. For example, an 8-bit accumulation might take only 0.03 pJ, while data movement on Loihi [4] costs 3-4 pJ/bit/hop. Could you please discuss your energy calculation a bit more instead of considering only MAC and AC? This will greatly enhance the credibility of the paper and make it more reasonable. I will raise my score if you provide these results and show that the SNN is still more energy-efficient than the ANN.
> >
> > [1] Dampfhoffer, Manon, et al. "Are SNNs really more energy-efficient than ANNs? An in-depth hardware-aware study." IEEE Transactions on Emerging Topics in Computational Intelligence 7.3 (2022): 731-741.
> > [2] Bhattacharjee, Abhiroop, et al. "Are SNNs truly energy-efficient?—A hardware perspective." ICASSP 2024-2024 IEEE International Conference on Acoustics, Speech and Signal Processing (ICASSP). IEEE, 2024.
> > [3]Yin, Ruokai, et al. "Loas: Fully temporal-parallel dataflow for dual-sparse spiking neural networks." 2024 57th IEEE/ACM International Symposium on Microarchitecture (MICRO). IEEE, 2024.
> > [4]Davies, Mike, et al. "Loihi: A neuromorphic manycore processor with on-chip learning." Ieee Micro 38.1 (2018): 82-99.

---

> ### Author Response · Authors · 2025-11-27
> **Response to wbNY - Part 1**
>
> Thank you for your insightful comments. Our initial energy estimation was based on an idealized scenario where an SNN-oriented neuromorphic chip could efficiently handle data movement and memory access . However, We acknowledge that current neuromorphic hardware has not yet broken free from the limitations of the Von Neumann architecture. As such, the articles you referenced still claim that memory access and data movement are the primary sources of energy consumption (which we refer to as the read/write cost), which reflects the current state of technology.
>
> In order to further validate the effectiveness of our approach on existing hardware, we have expanded our energy analysis to explicitly include the costs associated with read/write cost. This allows us to measure the energy overhead of our approach on chips that have not yet overcome the Von Neumann limitations.
>
> Specifically, for the calculation of energy consumed by reading and writing weights and activations, we refer to the energy estimation methods for both ANN and SNN models presented in [1]. We set the 32-bit read/write energy for weights and activations, $E_{read}$ and $E_{write}$, to 5 pJ and conducted a comparison of the energy consumption between ANN and our spiking LLM. The total energy equations for ANN and SNN are given by:
>
> $$
> \begin{aligned}
> E_{total}^{ANN}&= 2E_{read}\cdot Count_{read}+ E_{write}⋅Count_{write}+ E_{MAC}\cdot Count_{MAC}+E_{AC}\cdot Count_{AC} \\\\
> E_{total}^{SNN}&= E_{neuron}+ (1+1/32)E_{read}\cdot \sum_t Count_{read}^t+ 1/32 \cdot E_{write}⋅\sum_t Count_{write}^t+ E_{MAC}\cdot Count_{MAC}+E_{AC}\cdot Count_{AC}
> \end{aligned}
> $$
>
> The factors of $1/32$ in $E_{read}$ and $E_{write}$ in the SNN formula are due to the fact that activations in SNNs are represented using 1-bit spike.
>
> Table: Energy consumption of LLaMA-3-8B under ANN and our spiking LLMs, including both read and write operations.
> | Method | Read/Write Cost (J) | MAC & AC Cost (J) | Total Energy Cost (J) | Total Energy Cost Relative to ANN|
> |:-:|:-:|:-:|:-:|:-:|
> | ANN |38.47|18.00|56.47|100.00%|
> |Ours (Grain=2)|38.35|10.44|48.84|86.48%|
> |Ours (Grain=3)|38.93|10.46|49.43|87.53%|
>
>
> Table: Relative energy consumption of LLaMA-3-8B under ANN and our spiking LLMs, including both read and write operations.
> | Method | Proportion of Read/Write Cost| Proportion of MAC & AC Cost |
> |:-:|:-:|:-:|
> | ANN |68.12%|31.88%|
> |Ours (Grain=2)|78.52%|21.38%|
> |Ours (Grain=3)|78.76%|21.16%|
>
>
> As you correctly pointed out, memory access accounts for at least 68% of the total energy consumption in both ANN and current SNN models, making it the primary source of energy cost. Nevertheless, thanks to the sparsity inherent in SNN computations, our updated results show that the total energy consumption of our spiking LLMs remains over 12% lower than that of the ANN baseline.

---

> > ### Comment · Reviewer_wbNY · 2025-11-27
> >
> > Thanks for your response. It’s now much more reasonable to me. I will raise my scores to 6. Good luck!

---

> > > ### Author Response · Authors · 2025-11-27
> > > **Response to wbNY**
> > >
> > > We sincerely appreciate your time and effort in reviewing our work, and we are honored by your recognition of our research. Your valuable feedback is greatly appreciated.

---

> ### Author Response · Authors · 2025-11-27
> **Response to wbNY - Part 2**
>
> As correctly noted in the reviewer’s comment and supported by [2], the energy consumption of SNNs is closely tied to the spike firing rate. To address this issue, we propose a masking mechanism. Specifically, we exploit the characteristic of phase coding, where the encoding value decreases as the timestep increases. Consequently, spikes from neurons that fire early can be considered redundant, and those occurring at later timesteps can be discarded. This strategy effectively reduces the spike firing rate by eliminating redundant spikes without significantly impacting performance. As a result, the increased activation sparsity leads to a substantial reduction in the overall energy consumption of the SNN.
>
> Table: Energy Consumption of LLaMA-3-8B for ANN and our spiking LLMs with mask, including both read and write operations.
> | Method |Avg.ACC|PPL| Read/Write Cost (J) | MAC & AC Cost (J) | Total Energy Cost (J) | Total Energy Cost Relative to ANN|
> |:-:|:-:|:-:|:-:|:-:|:-:|:-:|
> | ANN |71.19|6.14|38.47|18.00|56.47|100.00%|
> |Ours (Grain=2)+Mask|66.18|8.82|34.02|6.08|40.13|71.06%|
> |Ours (Grain=3)+Mask|66.50|9.55|34.35|6.14|40.53|71.77%|
>
> Table: Relative energy consumption of LLaMA-3-8B under ANN and our spiking LLMs with mask, including both read and write operations.
> | Method | Proportion of Read/Write Cost| Proportion of MAC & AC Cost |
> |:-:|:-:|:-:|
> | ANN |68.12%|31.88%|
> |Ours (Grain=2)+Mask|84.77%|15.15%|
> |Ours (Grain=3)+Mask|84.75%|15.15%|
>
>
> In our energy estimation, we also include the cost of the masking operation. To be cautious, we upper-bound this cost by assigning the mask the same energy as a full neuron-level computation. While the energy consumption per neuron increases by a factor of two, this cost is more than compensated for by the lower spike firing rate resulting from the masking operation, ultimately yielding a lower value for $\tilde{Count}$ than for $Count$ in the energy calculation.
>
> $$
> \begin{aligned}
> E_{total}^{mask}=&  2\cdot E_{neuron}+ (1+1/32)E_{read}\cdot \sum_t \tilde{Count^t_{read}}+ \\\\\ &1/32 \cdot E_{write}⋅\sum_t \tilde{Count^t_{write}}+ E_{MAC}\cdot \tilde{Count_{MAC}}+E_{AC}\cdot \tilde{Count_{AC}}
> \end{aligned}
> $$
>
> Nevertheless, even under this assumption, the mask-related cost still accounts for only a small fraction of the total SNN energy, as neuron computation contributes relatively little compared with data movement and memory access. In contrast, the sparsity introduced by the mask significantly reduces both data movement and memory access, which dominate the overall energy budget.
>
> Of course, we believe that as SNN hardware designs continue to evolve, particularly with the rapid progress in compute-in-memory architectures, the energy consumption of SNNs will decrease even further.
>
> We believe your suggestion has greatly enhanced the rigor and completeness of our evaluation. Thank you once again for your valuable feedback, which has significantly improved the quality of this work. If you have any further concerns or questions, please feel free to raise them, and we will do our best to address them.
>
> [1] Hwang, Sangwoo, et al. "SpikedAttention: Training-free and fully spike-driven transformer-to-snn conversion with winner-oriented spike shift for softmax operation." Advances in Neural Information Processing Systems 37 (2024).
>
> [2] Dampfhoffer, Manon, et al. "Are SNNs really more energy-efficient than ANNs? An in-depth hardware-aware study." IEEE Transactions on Emerging Topics in Computational Intelligence 7.3 (2022): 731-741.

---

### Official Review · Reviewer_qqbp · 2025-10-30

**Soundness:** 2
**Presentation:** 2
**Contribution:** 2
**Rating:** 4
**Confidence:** 3

**Summary:**

The paper propose a ANN to SNN conversion method that accounts for the distribution of activation values.

**Strengths:**

The proposed method is shown to improve accuracy at a reasonable cost in the experiments.

**Weaknesses:**

The opening claim that activation distribution was not considered in previous work is clearly incorrect. The following are some examples that do just that - and not compared or even cited in the paper:

. Li, C., Ma, L., & Furber, S. (2022).
Quantization Framework for Fast Spiking Neural Networks (QFFS). Frontiers in Neuroscience.
Available at: https://www.frontiersin.org/articles/10.3389/fnins.2022.918793/full

2. Moser, B. A., & Lunglmayr, M. (2023).
Quantization in Spiking Neural Networks. arXiv preprint arXiv:2305.08012.
Available at: https://arxiv.org/abs/2305.08012

3. Guo, Y., et al. (2023).
RMP-Loss: Regularizing Membrane Potential Distribution for Spiking Neural Networks. Proceedings of the IEEE/CVF International Conference on Computer Vision (ICCV).
Available at: https://openaccess.thecvf.com/content/ICCV2023/papers/Guo_RMP-Loss_Regularizing_Membrane_Potential_Distribution_for_Spiking_Neural_Networks_ICCV_2023_paper.pdf

The time steps of 8 is also relatively large compared to recent work.

Overall, while I can see the approach seems intuitively on the right path, the evaluation needs to be improved.

**Questions:**

1. How does the time step size affect the conclusions?

2. Does it work for other models and data sets?

**Details Of Ethics Concerns:**

None.

---

> ### Author Response · Authors · 2025-11-21
> **Response to Reviewer qqbp - Part 1**
>
> Thank you to the reviewers for the energy and time dedicated to our paper. You can find the responses to all weaknesses and questions below. We have also made revisions to the original paper, with the modified parts highlighted in blue.
>
> **W1: The opening claim that activation distribution was not considered in previous work is clearly incorrect.**
>
> We thank the reviewer for pointing this out.
>
> * We agree that our original opening statement is misleading, and that prior works have indeed considered activation distribution in related contexts. We have revised the corresponding wording in the updated manuscript to avoid this misleading claim. The revised sentences are "Unfortunately, there exists the conversion error arising from distribution misalignment, which is a long-standing inherent problem in such conversions [1].
> However, current ANN-to-SNN conversion frameworks for LLM tend to overlook non-uniform activation distributions, leading to latent errors owing to distributional misalignment [2]." A more careful discussion of the novelty of our core problem and framework is provided in our response to W2 of Reviewer fm6A.
>
> * For the example papers you provided, we summarize them as follows:
>   * [3] introduces a novel method called QFFS (Quantization Framework for Fast Spiking Neural Networks), which combines quantization techniques from ANN with SNN to significantly reduce inference latency while maintaining high accuracy. This paper improves inference speed by applying low-bit precision activation quantization and addresses the issue of occasional noise in SNNs, but it does not explicitly discuss how the activation distribution affects low-bit quantization or occasional noise.
>
>   * [4] explores the LIF neuron model in the context of SNN and frames its operation as a quantization process. This paper introduces a novel reset-to-mod re-initialization variant, which is shown to preserve the quantization bound under general conditions, unlike the traditional reset-to-zero or reset-by-subtraction methods. It concludes by proposing a new error bound for SNNs based on this quantization perspective, but this bound is not derived by explicitly taking the activation distribution into account.
>
>     However, our understanding is that while these works [3,4] provide valuable exploration in related directions, they do not explicitly consider the conversion error induced by activation distributions as a core problem, as we do. Therefore, we believe that their focus and technical route differ from ours to some extent, and they are better viewed as complementary rather than directly competing approaches to our framework.
>
>   * [5] introduces RMP-Loss, a novel loss function designed to mitigate quantization errors in SNN. By regularizing the membrane potential distribution, the method adjusts the membrane potential to values closer to the 0/1 spike, thus reducing information loss caused by the quantization process.
>
>     In [5], the method of adjusting membrane potential distributions only partially alleviates the conversion error arising from distribution misalignment that we explicitly target in this work. Specifically, it merely encourages the membrane potential to be closer to values that trigger 0 or 1 spikes, but the resulting activation distribution after this adjustment remains non-uniform. Therefore, the issue of distribution misalignment remains unresolved. Overall, while the method in [5] may appear to touch upon activation distributions, the problem it addresses is not the same as ours, nor does it explicitly model or minimize the conversion error due to distribution misalignment, which is the core focus of our work. Finally, we note that the method in [5] requires retraining model weights, making it difficult to apply to LLMs with billions of parameters, whereas our approach avoids modifying the original model weights altogether.
>
>
> [1] Datta, Gourav, and Peter A. Beerel. "Can deep neural networks be converted to ultra low-latency spiking neural networks?." 2022 Design, Automation & Test in Europe Conference & Exhibition (DATE). IEEE, 2022.
>
> [2] Chen, Long, Xiaotian Song, and Yanan Sun. "LAS: Loss-less ANN-SNN Conversion for Fully Spike-Driven Large Language Models." arXiv preprint arXiv:2505.09659 (2025).
>
> [3] Li, Chen, Lei Ma, and Steve Furber. "Quantization framework for fast spiking neural networks." Frontiers in Neuroscience 16 (2022): 918793.
>
> [4] Moser, Bernhard A., and Michael Lunglmayr. “Quantization in Spiking Neural Networks.” arXiv preprint arXiv:2305.08012 (2023).
>
> [5] Guo, Yufei, et al. "Rmp-loss: Regularizing membrane potential distribution for spiking neural networks." Proceedings of the IEEE/CVF International Conference on Computer Vision. 2023.

---

> ### Author Response · Authors · 2025-11-21
> **Response to Reviewer qqbp - Part 2**
>
> **W2: The time steps of 8 is also relatively large compared to recent work.**
>
> We thank the reviewer for this comment and for drawing attention to the choice of timestep T.
>
> * First, we would like to clarify the context of our work. There are two mainstream routes to obtain SNNs:
>
>   * Direct training of SNNs, where the SNN is trained end-to-end with surrogate gradients.
>
>   * ANN-to-SNN conversion, where a pretrained ANN is converted into an SNN.
>
>   Our framework clearly belongs to the ANN-to-SNN conversion paradigm. In the direct-training setting, it is indeed common to use small timesteps, especially on relatively shallow networks [1-3]. In contrast, ANN-to-SNN conversion methods typically require much larger timesteps to faithfully approximate the ANN activations [4-6]. For LLM-scale models, directly training SNNs end-to-end is generally impractical due to optimization difficulty and computational cost, so ANN-to-SNN conversion is the realistic route. **Under this route, a timestep of T=8 is already quite small**.
>
> * Second, if we look specifically at existing spiking LLM works, our choice of T=8 is competitive:
>
>   * FAS (on 7B-scale models) uses 8-32 timesteps, and the performance degradation at T=8 is non-trivial [7].
>
>   * LAS uses 16 timesteps, which is 2×larger than our setting [8].
>
>   * SpikeGPT employs thousands of timesteps, which is orders of magnitude larger than our T=8 [9].
>
>   * SpikeLLM is the only methods as we know with relatively small timesteps, but it relies on a heavy post-conversion calibration and retraining process after conversion [10].
>
>   In contrast, our framework achieves strong performance at T=8 without modifying or retraining the LLM weights, by only optimizing neuron parameters via our distribution-aware multi-granularity phase coding. **Therefore, within the ANN-to-SNN LLM conversion regime, we believe that T=8 is not large, but rather a reasonably low timestep**.
>
> [1] Zhou, Zhaokun, et al. "Spikformer: When spiking neural network meets transformer." arXiv preprint arXiv:2209.15425 (2022).
>
> [2] Xing, Xingrun, et al. "Spikelm: Towards general spike-driven language modeling via elastic bi-spiking mechanisms." arXiv preprint arXiv:2406.03287 (2024).
>
> [3] Lv, Changze, et al. "Spikebert: A language spikformer trained with two-stage knowledge distillation from bert." (2023).
>
> [4] Zhao, Lusen, et al. "TTFSFormer: A TTFS-based Lossless Conversion of Spiking Transformer." Forty-second International Conference on Machine Learning.
>
> [5] You, Kang, et al. "Spikezip-tf: Conversion is all you need for transformer-based snn." arXiv preprint arXiv:2406.03470 (2024).
>
> [6] Deng, Shikuang, and Shi Gu. "Optimal conversion of conventional artificial neural networks to spiking neural networks." arXiv preprint arXiv:2103.00476 (2021).
>
> [7] Chen, Long, et al. "Fas: Fast ann-snn conversion for spiking large language models." arXiv preprint arXiv:2502.04405 (2025).
>
> [8] Chen, Long, Xiaotian Song, and Yanan Sun. "LAS: Loss-less ANN-SNN Conversion for Fully Spike-Driven Large Language Models." arXiv preprint arXiv:2505.09659 (2025).
>
> [9] Zhu, Rui-Jie, et al. "Spikegpt: Generative pre-trained language model with spiking neural networks." arXiv preprint arXiv:2302.13939 (2023).
>
> [10] Xing, Xingrun, et al. "SpikeLLM: Scaling up spiking neural network to large language models via saliency-based spiking." arXiv preprint arXiv:2407.04752 (2024).
>
> **Q1: How does the time step size affect the conclusions?**
>
> We thank the reviewer for this question. The effect of the timestep T on our framework can be understood from two perspectives:
>
> * **Effect of T on SNN performance via representational capacity**
>
>   In our phase-coding setting, a spiking neuron can represent up to $2^T$ distinct discrete values with T timesteps. So:
>
>   **Larger T** ⇒ **more representable discrete values** ⇒ **stronger approximation ability for the ANN activations**.
>
>   Consequently, the performance of the converted SNN improves as T increases, because the neuron can more finely approximate the underlying continuous activations. Our empirical results follow this behavior: when we increase T, the SNN's perplexity/accuracy improves for our framework.
>
> * **Effect of T on the role of granularity**
>
>   The second aspect is how T interacts with the number of granularities in our multi-granularity phase coding. **T controls how many discrete values are available in total, and the granularity design controls how these discrete values are distributed over the activation range**. A more detailed analysis of how timestep and granularity jointly affect performance and error is provided in our response to W4 of Reviewer fm6A.

---

> ### Author Response · Authors · 2025-11-21
> **Response to Reviewer qqbp - Part 3**
>
> **Q2: Does it work for other models and datasets?**
>
> We thank the reviewer for raising this question about the generalization of our framework. We agree that evaluating our framework beyond language models can further strengthen the empirical evidence for its effectiveness.
>
> * To this end, **we have extended our distribution-aware multi-granularity phase coding from LLMs to a multimodal model**, which is structurally and functionally different from language models. Due to the limited time available during the rebuttal period, we extend our framework to CLIP and evaluate the performance of the spiking CLIP model on image classification tasks.
>
> * **The results show that our framework exhibits strong scalability when applied to CLIP and its corresponding image datasets**. We have included these additional experiments in the revised version (Appendix D.4) to better demonstrate the generality of our approach across different models and modalities.
>
> Table: Performance Comparison Results on ImageNet, CIFAR10, and CIFAR100 using CLIP model. "FP32" represents the performance of the ANN evaluated under the float32 precision.
>
> | Model | Method | T | ImageNet | CIFAR10 | CIFAR100 | Avg. ACC |
> |-----------|-----------------|-----|----------|---------|----------|----------|
> | ViT-B/32  | FP32 | N/A | 57.71    | 89.69   | 64.01    | 70.47    |
> | ViT-B/32  | LAS | 8   | 55.42    | 89.27   | 66.22    | 70.30    |
> | ViT-B/32  | Ours (Grain=2)  | 8   | 56.72    | 90.48   | 66.11    | 71.10    |
> | ViT-B/32  | Ours (Grain=3)  | 8   | 56.78    | 90.23   | 65.64    | 70.88    |
> | ViT-B/16  | FP32 | N/A | 63.42    | 90.82   | 67.07    | 73.77    |
> | ViT-B/16  | LAS | 8   | 58.87    | 84.59   | 59.77    | 67.74    |
> | ViT-B/16  | Ours (Grain=2)  | 8   | 60.68    | 89.70   | 65.49    | 71.96    |
> | ViT-B/16  | Ours (Grain=3)  | 8   | 61.22    | 89.77   | 65.25    | 72.08    |
> | ViT-L/14  | FP32 | N/A | 71.13    | 95.82   | 76.41    | 81.12    |
> | ViT-L/14  | LAS | 8   | 69.71    | 88.82   | 70.29    | 76.27    |
> | ViT-L/14  | Ours (Grain=2)  | 8   | 69.61    | 94.99   | 77.61    | 80.74    |
> | ViT-L/14  | Ours (Grain=3)  | 8   | 69.63    | 94.87   | 77.17    | 80.56    |

---

> ### Author Response · Authors · 2025-11-27
> **Response to Reviewer qqbp**
>
> Dear Reviewer, thank you for taking the time to evaluate our work. To enhance the completeness of our submission, we proactively added additional experiments on Qwen2-7B using Multi-Granularity Phase Coding. **The results show that our method maintains high performance on Qwen2-7B, further demonstrating its effectiveness and scalability**.
>
> Table: Results on Qwen2-7B. "Grain" denotes the number of granularities. "PPL" denotes the perplexity on Wikitext2.
> | Method | T    | WinoGrande |  ArcC |  ArcE |  PiQA |  PPL |
> |:------:|:-:|:-:|:-:|:-:|:-:|:-:|
> | Qwen2-7B  | N/A | 72.38 | 49.91 | 74.71 | 81.23 | 7.14 |
> | LAS| 8 | 70.96 | 50.60 | 74.20 | 80.52 | 10.18|
> | SpikedAttention| 8 | 61.96 | 28.33 | 48.96 | 65.13 | >100 |
> | Ours (Grain=2) | 8 | 73.40 | 50.60 | 74.12 | 81.01 | 7.41 |
> | Ours (Grain=3) | 8 | 72.53 | 50.60 | 74.07 | 80.85 | 7.42 |

---

> ### Author Response · Authors · 2025-11-28
> **Response to Reviewer qqbp**
>
> We sincerely appreciate your thoughtful review and the insightful questions you raised. We would be grateful for any additional comments or suggestions you may have. We look forward to continuing the discussion and addressing the points you highlighted.

---

> ### Author Response · Authors · 2025-12-02
> **Response to Reviewer qqbp - Part 2: Additional Experiments**
>
> In response to the reviewer’s comment in W2, we conduct experiments with T = 6 on LLaMA-2-7B, and the results confirm that our approach achieves strong performance and can be applied to even lower timesteps.
>
> Table: Results on LLaMA-2-7B with $T=6$. "Grain" denotes the number of granularities. "PPL" denotes the perplexity on Wikitext2.
> | Method    | T | WinoGrande | ArcC  | ArcE  | PiQA  | Avg. Acc | PPL   |
> |------------------|---|------------|-------|-------|-------|----------|-----|
> | LLaMA-2-7B       | N/A | 69.06     | 46.33 | 74.54 | 79.05 | 67.25    | 5.47  |
> | LAS              | 6 | 67.96      | 44.28 | 72.52 | 77.86 | 65.65    | 45.50 |
> | SpikedAttention  | 6 | 66.69      | 41.64 | 70.03 | 76.77 | 63.78    | 50.05 |
> | Ours (Grain=2)   | 6 |  67.64| 45.31| 72.26|  77.58 | 65.70 |   12.19    |
> | Ours (Grain=3)   | 6 | 68.98      | 44.37 | 72.52 | 77.86 | 65.93    | 10.79 |

---

### Official Review · Reviewer_xCuD · 2025-11-01

**Soundness:** 3
**Presentation:** 3
**Contribution:** 3
**Rating:** 6
**Confidence:** 5

**Summary:**

This paper proposes an Alternating Optimization Neuron Training algorithm for directly training Spiking Large Language Models (Spiking LLMs) based on multi-granularity phase coding. The core innovation lies in decoupling the spiking threshold θ(t) and reset mechanism h(t) from the traditional decoding factor d(t), treating them as independent, learnable parameters. By alternately optimizing these learnable parameters with the network weights, the method aims to minimize information loss during the ANN-to-SNN conversion, thereby achieving high accuracy while significantly reducing energy consumption. The paper claims state-of-the-art results in both performance and energy efficiency.

**Strengths:**

1.The proposed "multi-granularity phase coding" framework and the idea of decoupling θ(t) and h(t) from d(t) are innovative.

2.The paper provides a convergence proof for the proposed alternating optimization algorithm.

**Weaknesses:**

1.The paper does not sufficiently demonstrate the necessity of the proposed decoupling mechanism. It lacks critical ablation studies to show how much performance would degrade if θ(t) and h(t) were not decouple.

2.The paper mentions a 42.0% energy reduction but does not specify the baseline or the evaluation methodology. Is it an estimate based on spike count in a simulator? Or is it actual power consumption measured on specific hardware?

**Questions:**

1.In the alternating optimization process, how are the convergence speed and stability of the two steps—optimizing weights B with fixed thresholds o, and optimizing thresholds o with fixed weights B? Is one step dominant? Have you tried joint optimization instead of alternating optimization?

2.How are the learnable bases B in "multi-granularity phase coding" initialized? What is their evolution trend during training?

3.The paper assumes a fixed total number of timesteps T. In practical inference, can true "event-driven" or "adaptive timestep" computation be achieved?

---

> ### Author Response · Authors · 2025-11-21
> **Response to Reviewer xCuD - Part 1**
>
> Thank you to the reviewers for the energy and time dedicated to our paper. You can find the responses to all weaknesses and questions below. We have also made revisions to the original paper, with the modified parts highlighted in blue.
>
> **W1: The paper does not sufficiently demonstrate the necessity of the proposed decoupling mechanism. It lacks critical ablation studies to show how much performance would degrade if $\theta(t)$ and $h(t)$ were not decouple.**
>
> We thank the reviewer for this comment and are happy to clarify the motivation and necessity of the proposed decoupling mechanism at two levels:
>
> * **Decoupling $\boldsymbol{\theta}$ and $\boldsymbol{h}$ from the bases $\boldsymbol{B}$ (why alternating optimization)**
>
>   In our framework, these parameters play different roles:
>     * The bases $\boldsymbol{B}$ determine the distribution of discrete representable values. This is exactly what our multi-granularity phase coding focuses on: by using multiple bases, we shape how discrete values are distributed to better match the activation distribution.
>
>     * The neuron parameters $\boldsymbol{\theta}$ and $\boldsymbol{h}$ determine how a given continuous input is mapped to one of those discrete values.
>
>   Therefore, the two processes are conceptually distinct:
>
>     * where the discrete values lie (controlled by $\boldsymbol{B}$), and
>
>     * how a continuous activation is mapped onto those values (controlled by $\boldsymbol{\theta}$ and $\boldsymbol{h}$).
>
>   Our decoupling mechanism explicitly separates them so that, in our alternating optimization:
>
>     * We optimize $\boldsymbol{B}$ (with multi-granularity design) to better match the activation distribution.
>
>     * Given a fixed discrete distribution,  we then optimize $\boldsymbol{\theta}$ and $\boldsymbol{h}$ to improve the mapping accuracy from continuous activations to those values.
>
>   This decouple is precisely what allows our framework to effectively minimize the conversion error arising from distribution misalignment, and is therefore necessary for our optimization procedure.
>
>   Moreover, from the perspective of the convergence of alternating optimization, the loss function is non-differentiable with respect to $\boldsymbol{h}$ and $\boldsymbol{\theta}$, so backpropagation for these variables must rely on surrogate gradients, whereas the optimization of $\boldsymbol{B}$ can directly use the true gradients. Therefore, intuitively, if $\boldsymbol{h}$, $\boldsymbol{\theta}$, and $\boldsymbol{B}$ are optimized simultaneously, the errors introduced by the surrogate gradients will propagate to the updates of $\boldsymbol{B}$, thereby amplifying the overall optimization error and reducing the stability of convergence.
>
>
> * **Decoupling $\boldsymbol{\theta}$ and $\boldsymbol{h}$ from each other (why $\boldsymbol{\theta}$≠$\boldsymbol{h}$)**
>
>   Beyond decoupling from $\boldsymbol{B}$, we also decouple the threshold $\boldsymbol{\theta}$ and the reset term $\boldsymbol{h}$ from each other. In conventional formulations, they are often tied, which reduces the degrees of freedom of the neuron dynamics. In our setting, once the discrete values (determined by $\boldsymbol{B}$) are fixed, the neuron still needs enough flexibility to shape the mapping from continuous activations to these values. By allowing $\boldsymbol{\theta}$ and $\boldsymbol{h}$ to vary independently, we:
>
>     * Increase the expressive power of the neuron dynamics;
>
>     * Allow more flexible control over when to fire (via $\boldsymbol{\theta}$) and how the membrane potential evolves after firing (via $\boldsymbol{h}$);
>
>     * Enable a finer adjustment of the mapping between continuous activations and discrete values.
>
>   Empirically, we observe that this extra flexibility helps reduce the approximation error between the SNN neuron output and the original ANN activation. We have clarified this design rationale in the revised version (Subsection Decoupling $\boldsymbol{h}$ and $\boldsymbol{\theta}$ from Each Other of Appendix D.2) and explicitly state that the decoupling of $\boldsymbol{\theta}$, $\boldsymbol{h}$, and $\boldsymbol{B}$ is necessary to (i) separately optimize discrete-level distribution and mapping, and (ii) provide sufficient flexibility in the neuron dynamics.
>
>   Table: Results on whether to decouple $\boldsymbol{h}$ and $\boldsymbol{\theta}$. "PPL" denotes the perplexity on Wikitext2.
>
>   |Model|T/Grain|Decouple|WinoGrande|ArcC|ArcE|PiQA|PPL|
>   |-|-|-|-|-:|-:|-:|-:|
>   | LLaMA-2-7B|8/2|No|70.24|45.65|74.03|77.86|7.62|
>   | LLaMA-2-7B|8/2|Yes|70.56|46.16|73.99|77.97|6.71|
>   | LLaMA-2-7B|8/3|No|70.17|45.90|73.86|77.97|6.90|
>   | LLaMA-2-7B|8/3|Yes|70.96|46.08|74.33|77.86|7.10|
>   | LLaMA-3-8B|6/2|No|71.98|45.90|71.80|75.03|8.00|
>   | LLaMA-3-8B|6/2|Yes|73.16|47.87|73.74|77.64|8.04|
>   | LLaMA-3-8B|6/3|No|73.40|47.78|72.26|75.35|7.63|
>   | LLaMA-3-8B|6/3|Yes|73.24|49.23|73.82|76.82|8.53|

---

> ### Author Response · Authors · 2025-11-21
> **Response to Reviewer xCuD - Part 2**
>
> **W2: The paper mentions a 42.0% energy reduction but does not specify the baseline or the evaluation methodology. Is it an estimate based on spike count in a simulator? Or is it actual power consumption measured on specific hardware?**
>
> We thank the reviewer for this important question regarding our energy reduction claim.
>
> To clarify, our energy evaluation approach is based on estimating the total number of MAC and AC operations required for inference, multiplied by their respective energy costs. Specifically, we calculate the average energy consumption per sample during inference by collecting the number of MAC and AC operations performed, and then applying the corresponding energy-per-operation values. This provides an estimate of the average energy per sample during inference, based on the number of operations.
>
> * We acknowledge that the energy reduction presented in the paper could have been described more clearly. Our results are based on operation counts rather than direct measurements of power consumption on specific hardware.
>
> * We acknowledge that the description and conclusions in our paper regarding energy are somewhat misleading. For further clarification, we refer to our response to W1 of Reviewer fm6A, where we discuss this issue, as well as the reasonableness of this energy estimation approach.
>
> Thank you for pointing this out, and we have made the necessary revisions to improve the clarity of this section in the updated version.
>
> **Q1: In the alternating optimization process, how are the convergence speed and stability of the two steps—optimizing weights B with fixed thresholds o, and optimizing thresholds o with fixed weights B? Is one step dominant? Have you tried joint optimization instead of alternating optimization?**
>
> We thank the reviewer for this insightful question about the convergence behavior of our alternating optimization scheme.
>
> * In response, we have added the training loss curves for the two steps in the revised version (Subsection Interaction between the Timestep T and the Number of Granularities. of Appendix D.2). From these curves, we observe that the training loss decreases in a stable manner and approaches convergence. Moreover, the two stages in the alternating optimization contribute comparably to the loss reduction, indicating that neither phase is dominant in the optimization process.
>
>
> * We have also followed the reviewer's suggestion and experimented with joint optimization. In this setting, we observed a worse downstream performance. We believe this is because, without decoupling the two stages, updates to $\boldsymbol{B}$ and {$\boldsymbol{\theta}$, $\boldsymbol{h}$} interfere with each other: modifying the discrete values distribution (via $\boldsymbol{B}$) and the mapping (via $\boldsymbol{\theta}$, $\boldsymbol{h}$) at the same time makes the optimization landscape harder to navigate. For the necessity of alternating optimization, please refer to our response to W1.
>
>   Table: Results on joint optimization and alternating optimization. "PPL" denotes the perplexity on Wikitext2.
>
>   | Model | T/Grain | Method | WinoGrande | ArcC | ArcE | PiQA |  PPL |
>   |------------|---------|--------|------------|------:|------:|------:|-----:|
>   | LLaMA-2-7B | 8/2 | Joint  | 70.09 | 45.22 | 73.99 | 77.86 | 6.58 |
>   | LLaMA-2-7B | 8/2 | Alter  | 70.56 | 46.16 | 73.99 | 77.97 | 6.71 |
>   | LLaMA-2-7B | 8/3 | Joint  | 69.85 | 45.65 | 73.78 | 77.69 | 6.41 |
>   | LLaMA-2-7B | 8/3 | Alter  | 70.96 | 46.08 | 74.33 | 77.86 | 7.10 |
>   | LLaMA-3-8B | 6/2 | Joint  | 72.38 | 47.18 | 71.38 | 76.55 | 7.61 |
>   | LLaMA-3-8B | 6/2 | Alter  | 73.16 | 47.87 | 73.74 | 77.64 | 8.04 |
>   | LLaMA-3-8B | 6/3 | Joint  | 71.67 | 47.18 | 72.39 | 74.54 | 7.52 |
>   | LLaMA-3-8B | 6/3 | Alter  | 73.24 | 49.23 | 73.82 | 76.82 | 8.53 |
>
>
>   These new results have been added to the revised version (Subsection Comparison between Joint Optimization and Alternating Optimization of Appendix D.2), and we highlight that alternating optimization is chosen not only for conceptual clarity but also because it empirically achieves better performance than joint optimization in our setting.

---

> ### Author Response · Authors · 2025-11-21
> **Response to Reviewer xCuD - Part 3**
>
> **Q2: How are the learnable bases B in "multi-granularity phase coding" initialized? What is their evolution trend during training?**
>
> We thank the reviewer for this thoughtful question.
>
>   * **Initialization of the bases $\boldsymbol{B}$**
>
>     In our implementation, all learnable bases $\boldsymbol{B}$ in the multi-granularity phase coding are initialized to 2. This induces a uniform distribution of discrete points at the beginning of training. In other words, the initial state of our neuron behaves like a conventional phase-coding neuron, and the multi-granularity behavior is gradually learned.
>
>   * **Evolution trend during training**
>
>     To better illustrate how the bases evolve, we have added plots in the revised version (Subsection Interaction between the Timestep T and the Number of Granularities. of Appendix D.2) showing the trajectory of the bases $\boldsymbol{B}$ together with the training loss during the alternating optimization process. These curves clearly reflect the evolution of bases $\boldsymbol{B}$ and the training loss throughout the optimization process.
>
>     <!-- [TODO: exp trajectory of the bases $\boldsymbol{B}$ together with the training loss]. -->
>
> **Q3: The paper assumes a fixed total number of timesteps T. In practical inference, can true "event-driven" or "adaptive timestep" computation be achieved?**
>
> We thank the reviewer for this insightful question and are happy to clarify the relationship between our framework and event-driven as well as adaptive timestep.
>
> * **For "event-driven"**
>
>   Our framework is inherently event-driven at inference time. The proposed spiking neurons follow standard SNN dynamics: they only emit spikes when the membrane potential crosses the threshold, and downstream computation (AC operations) is triggered only by these spikes. As a result, when activations are sparse, many neurons remain silent and do not contribute any operations, which is exactly the event-driven behavior exploited in neuromorphic hardware. Therefore, although we assume a fixed total timestep T, the actual computation is event-driven.
>
> * **For "adaptive timestep"**
>
>   We apologize for the confusion caused by our wording regarding "adaptive". In our work, T is fixed, and what we adapt is not the total number of timesteps during inference, but rather how the T timesteps are allocated across different granularities in the multi-granularity phase coding. Concretely, given a fixed total T, our framework learns how to split these T steps into segments with different bases (granularities), i.e., how many steps use each base $B_i$.
>
>   This allocation is fully determined during the optimization stage and remains fixed during inference. Thus, there is no runtime adaptive change of the total timestep per token, per sample, or per layer in our current framework. The adaptivity we refer to is in the timestep allocation, not in dynamically changing T at inference time.

---

### Official Review · Reviewer_fm6A · 2025-11-03

**Soundness:** 3
**Presentation:** 3
**Contribution:** 3
**Rating:** 6
**Confidence:** 4

**Summary:**

To address the issue of mismatched non-uniform activation value distribution when converting LLM to SNN, this paper introduced Distribution-Aware Multi-Granularity Phase Coding with multiple learnable bases and proposed the Alternating Optimization Neuron Training Algorithm to optimize the parameters of SNN neurons, thereby reducing the conversion time.

**Strengths:**

The structure and writing of this paper are good and easy to understand. In terms of reducing conversion time, it improves efficiency and may be a promising method.

**Weaknesses:**

1. This paper claims that spiking large language models (LLMs) have significant advantages on neuromorphic hardware, but the final conclusion is that the energy consumption is reduced by 42% on GPUs. Moreover, the way energy consumption is presented is extremely unscientific and rigorous, which seriously undermines the technical contribution and persuasiveness of the paper. First, the paper cites Horowitz's 2014 proposal that "1 MAC ≈ 4.6 pJ and 1 AC ≈ 0.9 pJ under 45 nm CMOS technology" as the basis for energy consumption conversion, but there is no specific model of "commodity GPUs". The energy consumption designs of different GPUs vary greatly. The mainstream process of current "commercial GPUs" has entered 3~8 nm, and there is a technical generation gap of more than several decades between the two. The deviation between the calculation results and the actual energy consumption of the hardware is at least an order of magnitude. The methodology of estimating the actual energy consumption of contemporary GPUs using the energy consumption coefficient of 45 nm is completely invalid [1][2]. Secondly, the energy consumption of LLM does not only come from "computational operations (MAC/AC)". In contemporary GPUs, the energy consumption of memory access (i.e., HBM, SRAM read and write), data movement (data transmission in PCIe), and control unit scheduling often exceeds the computational energy consumption. Moreover, LLM inference is memory bound, and memory energy consumption accounts for the vast majority of the total energy consumption. The paper only estimates the total energy consumption based on the number of MAC/AC, completely ignoring key energy consumption items such as memory access and data movement, which has no correlation with the "end-to-end energy consumption" of the actual GPU operation.

2. The core problem pointed out by the authors of this paper is not new: conversion error arising from distribution misalignment. This problem does not only exist in the conversion of LLM to SNN, but is a long-standing inherent problem [3], and it still exists in the conversion of CNN to SNN.

3. The Distribution-Aware Multi-Granularity Phase Coding in this paper actually adjusts the parameters such as h(t) and d(t) in FS neurons to multi-granularity learnable parameters, while the parameters of the original FS neurons are also learnable and adjustable, and there is no essential difference [4].

4. The ablation study in Table 5 shows that when the granularity increases from 1 to 2 or 3, the performance gain is very limited, which means that the core contribution of the paper is actually the learnable B, rather than "multi-granularity". Compared with a carefully optimized "single-granularity distribution-aware phase coding", what is the real advantage of multi-granularity? It can be seen from Tables 1, 2, and 6 that when the time step is large (T=10), the performance gain by increasing the Grain is negligible, while when the time step is slightly smaller (T=8), the change in the result by changing the grain is slightly larger. Does this mean that the time step T is also an influencing factor? Therefore, an ablation experiment of time step T and grain should be added.

5. The meaning of the formula is ambiguous or even ambiguous: In formula (3), the author writes "Typically, $h(t)$, $d(t)$, and $\theta(t)$ are specified as $B^{-t}$ as follows:$v(t + 1) = v(t) - B^{-t}s(t),\ O(t) = B^{-t}s(t),\ s(t) = \Theta\left(v(t)-B^{-t}\right).$" But in formula (8), it is claimed that "we denote $\{h(t)\}_{t=1}^T$, $\{\theta(t)\}_{t=1}^T$, $\{d(t)\}_{t=1}^T$, $\{B_i\}_{i=1}^n$ as $h$, $\theta$, $d$, and $B$" So what is the final expression?

6. There is a lack of comparison with existing works. Only SpikeLLM, SpikedAttention, and LAS are compared, and more extensive control experiments are lacking.

[1] https://www.nvidia.com/content/dam/en-zz/Solutions/Data-Center/a100/pdf/nvidia-a100-datasheet-nvidia-us-2188504-web.pdf?utm_source=chatgpt.com

[2] https://resources.nvidia.com/en-us-gpu-resources/h100-datasheet-24306?utm_source=chatgpt.com

[3] Can Deep Neural Networks be Converted to Ultra  Low-Latency Spiking Neural Networks

[4] Optimized spiking neurons can classify images with high accuracy through temporal coding with two spikes

**Questions:**

The problem is as described above.

---

> ### Author Response · Authors · 2025-11-21
> **Response to Reviewer fm6A - Part 1**
>
> Thank you to the reviewers for the energy and time dedicated to our paper. You can find the responses to all weaknesses and questions below. We have also made revisions to the original paper, with the modified parts highlighted in blue.
>
> **W1: The energy consumption designs of different GPUs vary greatly.  The methodology of estimating the actual energy consumption of contemporary GPUs using the energy consumption coefficient of 45 nm is completely invalid. The paper only estimates the total energy consumption based on the number of MAC/AC, completely ignoring key energy consumption items such as memory access and data movement, which has no correlation with the "end-to-end energy consumption" of the actual GPU operation.**
>
> We thank the reviewer for their detailed comment and insightful points regarding our energy consumption estimation approach. We would like to address the concerns and clarify our framework.
>
> * **Clarification of Energy Consumption Terminology and Acknowledgment of Misleading Statements**
>
>   We acknowledge that the descriptions and conclusions in our paper regarding energy consumption are somewhat misleading (for example, we should not have used the term "end-to-end energy consumption"). We have corrected these misleading descriptions in the revised version. The revised sentences are in Section 6.3.
>
> * **Context of Energy Consumption in Brain-Inspired Hardware**
>
>   Thank the reviewer for correctly pointing out that modern GPUs vary greatly in energy-consumption design, and that memory access and data movement are key components that we did not explicitly account for in our original analysis. Existing LLM are implicitly designed for typical von Neumann architectures, where computation and storage are separated without considering the compute-in-memory characteristics of non-von Neumann neuromorphic chips [1,2].
>
>   By contrast, **our framework is intended as an architecture design for neuromorphic (non-von Neumann) chips.** Such brain-inspired hardware adopts compute-in-memory and event-driven execution, so the data representation and operator execution style are fundamentally different from those of conventional LLMs on GPUs. In this setting, MAC/AC operations are the primary contributors to energy consumption, while memory access and data movement are no longer the dominant terms.
>
>
> * **Mainstream Approach of MAC/AC Energy Consumption**
>
>   Regarding the approach used to estimate energy consumption for MAC/AC operations, we want to emphasize that this is the mainstream approach in energy estimation for SNN, as widely accepted in the field, and the energy-per-operation for MAC/AC is typically derived from standards based on 45 nm technology [3-10]. While this does not capture every aspect of energy consumption, it provides a reasonable approximation.
>
> * **Extended Energy Analysis with Explicit Read/Write Cost Modeling**
>
>   **To further validate the energy advantage of SNNs under data movement and memory access**, which we refer to as read and write cost, we extend our energy accounting to explicitly include these costs. Using the original SNN formulation, we observe more than a **12** percent energy reduction compared to ANN. With our proposed spike firing rate control mechanism, and after accounting for the additional energy introduced by the rate control itself, the energy consumption of our SNNs is further reduced to over **28** percent. Detailed calculations and discussion are provided in our response to reviewer **wbNY**.
>
>
> We hope these clarifications help to address the reviewer's concerns. Thank you again for your valuable feedback.
>
> [1] Muir, Dylan Richard, and Sadique Sheik. "The road to commercial success for neuromorphic technologies." Nature communications 16.1 (2025).
>
> [2] Wan, Weier, et al. "A compute-in-memory chip based on resistive random-access memory." Nature 608.7923 (2022).
>
> [3] Rathi, Nitin, and Kaushik Roy. "Diet-snn: Direct input encoding with leakage and threshold optimization in deep spiking neural networks." arXiv preprint (2020).
>
> [4] Li, Yuhang, et al. "Differentiable spike: Rethinking gradient-descent for training spiking neural networks." NeurIPS 34 (2021).
>
> [5] Zhou, Zhaokun, et al. "Spikformer: When spiking neural network meets transformer." arXiv preprint (2022).
>
> [6] Deng, Shikuang, et al. "Spiking Token Mixer: An event-driven friendly Former structure for spiking neural networks." Advances in Neural Information Processing Systems 37 (2024).
>
> [7] Zhao, Lusen, et al. "TTFSFormer: A TTFS-based Lossless Conversion of Spiking Transformer." ICML.
>
> [8] Wang, Yuchen, et al. "Signed Neuron with Memory: Towards Simple, Accurate and High-Efficient ANN-SNN Conversion." IJCAI. 2022.
>
> [9] Chen, Long, et al. "Fas: Fast ann-snn conversion for spiking large language models." arXiv preprint (2025).
>
> [10] Chen, Long, Xiaotian Song, and Yanan Sun. "LAS: Loss-less ANN-SNN Conversion for Fully Spike-Driven Large Language Models." arXiv preprint (2025).

---

> ### Author Response · Authors · 2025-11-21
> **Response to Reviewer fm6A - Part 2**
>
> **W2: The core problem pointed out by the authors of this paper is not new: conversion error arising from distribution misalignment. This problem does not only exist in the conversion of LLM to SNN, but is a long-standing inherent problem, and it still exists in the conversion of CNN to SNN.**
>
> We appreciate the reviewer's comment regarding the long-standing issue of conversion error arising from distribution misalignment.
> We agree that this problem is not new,  as it has already been pointed out in the context of CNN to SNN conversion.
> However, we believe that while this problem itself is not new, the framework we propose to address it in the context of LLMs is entirely novel, and there is significant innovation in how our framework solves it.
>
> * **Novel Background (Spiking Large Language Model) Brings New Challenges.** Under highly diverse activation distributions and with retraining being exceedingly difficult, the limited representational capacity of neurons at small time steps is further exacerbated by unacceptable error accumulation, posing a fundamental challenge. Specifically, the increased number of parameters in spiking LLMs significantly raises the computational cost of both forward and backward propagation, making retraining-based methods for aligning activation distributions with the neuronal representational range practically infeasible. Moreover, different spiking model architectures can lead to substantial discrepancies in activation distributions. For example, the distribution after softmax may differ drastically from those of the query, key, and value vectors, forcing neurons to fit these diverse activation distributions. Furthermore, as the network depth increases, the ANN-to-SNN conversion errors accumulate rapidly, leading to a significant degradation in accuracy. Under such challenges, to the best of our knowledge, we are the first to systematically identify and address conversion errors caused by distribution misalignment in the context of spiking LLMs.
>
> * **Innovative Framework: Alternating Optimization Neuron Training Algorithm.**  In our conversion framework, the alternating optimization algorithm we proposed minimizes the conversion error arising from distribution misalignment by alternately optimizing the neuron parameters (the necessity of alternating optimization is discussed in the response to W1 of Reviewer xCuD), which is a key innovation of our method. In addition, by training on pre-sampled activations, our method avoids the heavy cost of forward and backward propagation. The neuron-wise training procedure allows neurons to adapt to the diverse activation distributions arising from different components of the network. Reducing the conversion error arising from distribution misalignment effectively mitigates error accumulation in spiking LLMs. By doing so, we are able to perform the SNN conversion efficiently, without the need to modify the original LLM weights, resulting in significantly reduced computational overhead while improving conversion accuracy.
> * **Problem-Solving Perspective.** Reducing ANN-to-SNN conversion error caused by distribution misalignment can substantially improve SNN performance. Unlike existing methods [1,2], which adjust uniform quantization intervals via shifting and scaling to fit activation distributions, these approaches are inherently constrained by their fixed interval structure and, while effective for small models, do not transfer well to large-scale models.
> In contrast, our approach constructs non-uniform quantization intervals that more closely match the actual activation patterns. This introduces additional degrees of freedom, which is particularly beneficial for handling the skewed activation distributions commonly observed in spiking LLMs, thereby achieving superior performance at the LLM level.
>
>
> [1] Datta, Gourav, and Peter A. Beerel. "Can deep neural networks be converted to ultra low-latency spiking neural networks?." 2022 Design, Automation & Test in Europe Conference & Exhibition (DATE). IEEE, 2022.
>
> [2] Bojkovic, Velibor, et al. "Data driven threshold and potential initialization for spiking neural networks." International Conference on Artificial Intelligence and Statistics. PMLR, 2024.

---

> ### Author Response · Authors · 2025-11-21
> **Response to Reviewer fm6A - Part 3**
>
> **W3: The Distribution-Aware Multi-Granularity Phase Coding in this paper actually adjusts the parameters such as $h(t)$ and $d(t)$ in FS neurons to multi-granularity learnable parameters, while the parameters of the original FS neurons are also learnable and adjustable, and there is no essential difference.**
>
> We appreciate the reviewer's comment regarding the comparison between our Distribution-Aware Multi-Granularity Phase Coding and the standard FS neurons with learnable parameters.
>
> While it is true that our Distribution-Aware Multi-Granularity Phase Coding is built upon the foundation of FS neurons, we respectfully disagree that this means our proposed phase coding scheme lacks significance.
>
>
>
> FS neurons indeed provide a general and flexible framework in which parameters such as $h(t)$ and $d(t)$ are learnable and adjustable, resulting in a large search space. This generality presents both advantages and challenges: the flexibility of FS enables it to cover a broader solution space, which should theoretically lead to better outcomes; however, the vast non-convex solution space also makes the optimization process highly prone to falling into local optima, making it difficult to converge to an ideal solution in practical applications.
>
> Our key contribution lies in the addition of phase constraints, which narrows the search space and makes the optimization process more manageable. This reduction in complexity is not arbitrary. We can empirically validate this point through our ablation study, where we compare the performance of the model with the number of granularities set to T. In this setting, $\boldsymbol{d}$ has T learnable parameters (one for each timestep), which is essentially equivalent to the FS neuron expression. However, in this case, we observe that performance becomes suboptimal. This suggests that restricting the space of possible solutions (through our multi-granularity approach) indeed helps to achieve better performance by simplifying the optimization process.
>
> In summary, while the FS neuron framework provides a general basis, our framework of adding phase constraints and introducing multi-granularity brings improvements, as demonstrated by empirical results.
>
> Table: Ablation study on the number of granularities with T=8.
> | Models | Grain | Avg. ACC | Avg. PPL |
> |---------------|-------|----------|----------|
> | LLaMA-2-7B | 1 | 67.03 | 7.01 |
> | LLaMA-2-7B | 2 | 67.17 | 7.16 |
> | LLaMA-2-7B | 3 | 67.31 | 7.68 |
> | LLaMA-2-7B | T | 67.14 | 7.44 |
> | LLaMA-3-8B | 1 | 71.09 | 7.24 |
> | LLaMA-3-8B | 2 | 71.32 | 7.22 |
> | LLaMA-3-8B | 3 | 71.48 | 7.28 |
> | LLaMA-3-8B | T | 71.40 | 7.27 |

---

> ### Author Response · Authors · 2025-11-21
> **Response to Reviewer fm6A - Part 4**
>
> **W4: The ablation study in Table 5 shows that when the granularity increases from 1 to 2 or 3, the performance gain is very limited, which means that the core contribution of the paper is actually the learnable B, rather than "multi-granularity". Compared with a carefully optimized "single-granularity distribution-aware phase coding", what is the real advantage of multi-granularity? It can be seen from Tables 1, 2, and 6 that when the time step is large (T=10), the performance gain by increasing the Grain is negligible, while when the time step is slightly smaller (T=8), the change in the result by changing the grain is slightly larger. Does this mean that the time step T is also an influencing factor? Therefore, an ablation experiment of time step T and grain should be added.**
>
> We thank the reviewer for this careful analysis and for pointing out the interaction between the timestep T and the number of granularities (Grain). We address (1) the advantage of multi-granularity, and (2) the influence of T, including additional ablations.
>
> * **The advantage of multi-granularity**
>
>   We agree that learnable $\boldsymbol{B}$ is an essential part of our method. Even with Grain=1, making $\boldsymbol{B}$ learnable already yields improvements over conventional fixed-base phase coding (LAS and SpikeAttention). However, multi-granularity is not redundant with learnable $\boldsymbol{B}$. Instead, it strictly extends the solution space:
>
>   A single-granularity distribution-aware phase coding is a special case of our framework, obtained when all timesteps share the same base (Grain=1). When we increase the number of granularities, the model can non-uniformly allocate discrete values more flexibly. The solution space with Grain=2 or 3 strictly contains the Grain=1 solution space, so, in principle, more granularities can only help.
>
>   However, in practice, this larger solution space also makes optimization more non-convex and prone to local minima. Our experiments in response to W3 confirm exactly this trade-off. When we push granularity to the extreme (e.g., setting the number of granularities equal to the timestep T), the performance actually is not the best, indicating that excessive granularity makes the optimization harder and the solution is more likely to be suboptimal.
>
>   **Therefore, we do not claim that more granularities will always be better. Instead, we show that moderate multi-granularity provides improvements over the single-granularity case at the same T, while large numbers of granularities can hurt due to optimization difficulty.**
>
> * **Additional ablations**
>
>   We agree with the reviewer that the effect of Grain depends on the timestep T, and we appreciate the suggestion to study this more systematically.
>
>   Conceptually, in our phase-coding neuron, the number of representable discrete values grows as $2^T$. When T is large, the discrete representation is already quite dense, so redistributing these discrete values via multi-granularity (Grain>1) provides smaller gains. This explains why, in Tables 1, 2, and 9 (Table 6 in original paper), the improvement from increasing Grain at a larger T appears modest.
>
>   In contrast, when T is lower, the total number of discrete values is more limited, so where these values are placed becomes much more critical. In this regime, multi-granularity can reduce conversion error by allocating more resolution to high-density regions of the activation distribution. This matches the reviewer's observation that the gain from increasing Grain is more pronounced at a lower T.
>
>   To address the reviewer's request, we have added a more detailed ablation over both T and Grain in the revised version (Subsection Interaction between the Timestep T and the Number of Granularities of Appendix D.2). Specifically, we provide training loss curves for different combinations of T and Grain, showing how the optimization converges.

---

> ### Author Response · Authors · 2025-11-21
> **Response to Reviewer fm6A - Part 5**
>
> **W5: The meaning of the formula is ambiguous or even ambiguous**
>
> We thank the reviewer for pointing out the ambiguity in the formulation of Eqs. (3) and (8), and we apologize for the confusion caused by our presentation. We have revised the manuscript to make the relationship between these equations explicit.
>
> In our current formulation, Eqs.(1)-(3) are intended to describe the standard biophysical dynamics of conventional phase-coding neurons:
> * Eqs.(1)-(2) define the membrane potential update and spike generation process for a generic phase-coding neuron.
> * **Eq.(3) specifically corresponds to the conventional phase coding setting**, where the neuron parameters are typically tied to a single base B. This equation is meant to show the classical choice used in conventional phase coding
>
> By contrast, **Eq.(8) is the dynamic process of our proposed distribution-aware multi-granularity phase coding**. It is obtained by extending the conventional formulation in Eq.(3):
>
> * Starting from the standard form in Eq.(3) with a single base $B$. Eq. (8) introduces our multi-granularity design, in which the neuron parameters are no longer constrained to a single base, but instead are constructed from multiple bases. **This generalizes the conventional single-base scheme in Eq.(3)**.
>
> In the revised version (Section 3 and Section 4.2), we have clarified in the text that Eq.(8) is obtained by introducing our multi-granularity design to Eq.(3). This should remove the ambiguity and make the logical connection between Eq.(3) and Eq.(8) clear.

---

> ### Author Response · Authors · 2025-11-21
> **Response to Reviewer fm6A - Part 6**
>
> **W6: There is a lack of comparison with existing works. Only SpikeLLM, SpikedAttention, and LAS are compared, and more extensive control experiments are lacking.**
>
> We thank the reviewer for pointing out the importance of comprehensive comparisons. We would like to clarify the current state of the field and our choices of baselines.
> * First, **research on spiking LLMs is still at a very early stage**. To the best of our knowledge, there are currently very few works that explicitly target LLM-level spiking models. Among them, SpikeLLM and LAS are the representative methods that are designed for spiking LLMs and provide enough implementation details or code to enable a fair comparison. We have included both of them as baselines in our experiments.
>
> * In addition, we also compare with SpikedAttention, which is not a spiking LLM method but a phase-coding-based design for the attention module. We include SpikedAttention because it is one of **the closest existing works on phase coding for transformer**, and thus serves as a strong and relevant reference point for our framework, even though it is not a spiking framework designed for LLM. Beyond these, most prior SNN conversion methods either focus on Relu-based model or are not directly applicable to transformer-based LLMs, and therefore cannot be straightforwardly adapted to the spiking LLM.
>
> * Despite these limitations, we additionally implement **a temporal-coding spiking LLM based on the temporal coding** in [1] within our own framework to serve as an extra baseline. This allows us to directly compare our distribution-aware multi-granularity phase coding against a temporal coding scheme. The results are in the table below and we also have added them to the revised version (Section 6).
>
> Table: Results on LLaMA-2-7B. (Comparsion with TTFSFormer.)
> | PPL Perf. ↓ | T | Time Cost | Wikitext2 | C4 | Redpajama | Pile | Avg. PPL |
> |-----------------|-----|-----------|-----------|-------|-----------|------|---------|
> | TTFSFormer | 128 | N/A | 11.88 | 16.47 | 13.18 | 9.32 | 12.71 |
> | Ours (Grain=2) | 8 | 2m 01s | 6.71 | 8.96 | 7.23 | 5.74 | 7.16 |
> | Ours (Grain=3) | 8 | 2m 04s | 7.10 | 9.71 | 7.80 | 6.09 | 7.68 |
> | Ours (Grain=2) | 10 | 2m 25s | 5.50 | 7.05 | 5.68 | 4.67 | 5.73 |
> | Ours (Grain=3) | 10 | 2m 27s | 5.53 | 7.06 | 5.69 | 4.68 | 5.74 |
> | ACC Perf. ↑ | T | Time Cost | WinoGrande | ArcC | ArcE | PiQA | Avg. ACC |
> | TTFSFormer | 128 | N/A | 70.56 | 44.88 | 73.11 | 78.94 | 66.87 |
> | Ours (Grain=2) | 8 | 2m 01s | 70.56 | 46.16 | 73.99 | 77.97 | 67.17 |
> | Ours (Grain=3) | 8 | 2m 04s | 70.96 | 46.08 | 74.33 | 77.86 | 67.31 |
> | Ours (Grain=2) | 10 | 2m 25s | 70.48 | 46.50 | 73.91 | 78.29 | 67.30 |
> | Ours (Grain=3) | 10 | 2m 27s | 70.88 | 46.16 | 73.78 | 78.13 | 67.24 |
>
> Table: Results on LLaMA-3-8B. (Comparsion with TTFSFormer.)
> | PPL Perf. ↓ | T | Time Cost | Wikitext2 | C4 | Redpajama | Pile | Avg. PPL |
> |-----------------|-----|-----------|-----------|-------|-----------|------|---------|
> | TTFSFormer | 128 | N/A | 6.72 | 9.82 | 8.12 | 6.03 | 7.67 |
> | Ours (Grain=2) | 6 | 1m 36s | 8.04 |12.25 |10.14 | 7.23 | 9.42 |
> | Ours (Grain=3) | 6 | 1m 35s | 8.53 |13.18 |10.85 | 7.73 | 10.07 |
> | Ours (Grain=2) | 8 | 2m 01s | 6.32 | 9.14 | 7.69 | 5.74 | 7.22 |
> | Ours (Grain=3) | 8 | 1m 58s | 6.37 | 9.22 | 7.73 | 5.79 | 7.28 |
> | ACC Perf. ↑ | T | Time Cost | WinoGrande | ArcC  | ArcE | PiQA | Avg. ACC |
> | TTFSFormer | 128 | N/A | 72.69 | 52.90 | 77.65 | 79.33 | 70.64 |
> | Ours (Grain=2) | 6 | 1m 36s | 73.16 | 47.87 | 73.74| 77.64 | 68.10 |
> | Ours (Grain=3) | 6 | 1m 35s | 73.24 | 49.23 | 73.82| 76.82 | 68.28 |
> | Ours (Grain=2) | 8 | 2m 01s | 72.69 | 54.35 | 78.11| 80.14 | 71.32 |
> | Ours (Grain=3) | 8 | 1m 58s | 73.09 | 54.10 | 78.41| 80.30 | 71.48 |
>
>
>
> [1] Zhao, Lusen, et al. "TTFSFormer: A TTFS-based Lossless Conversion of Spiking Transformer." Forty-second International Conference on Machine Learning.

---

### Author Response · Authors · 2025-12-01
**Global Response to Reviewers and Area Chairs**

Dear Reviewer, Area Chair, Senior Area Chair, and Program Chair,
Thank you for your hard work and commitment during this unusual period.
## Summary of the Current Review Status
To facilitate your review, we have summarized the current status of the paper's evaluation as follows:

* In their feedback, reviewer wbNY clearly stated: *"Thanks for your response. It’s now much more reasonable to me. I will raise my scores to 6. Good luck!"*

* For reviewer qqbp, who has not yet had the chance to respond to our replies and whose score remains the only negative one, we appreciate their positive assessment of our work: *"Overall, while I can see the approach seems intuitively on the right path, the evaluation needs to be improved."* We have provided comprehensive responses to all the weaknesses and questions raised, and we believe that all concerns have been fully addressed.

* Lastly, reviewers fm6A, xCuD, and qqbp have not engaged in further interaction with us.

## Summary of the Key Issues and Solutions

Furthermore, to facilitate your review of our responses, we have summarized the key issues and outlined the solutions implemented.

1. **Explanation and Extension of Energy Consumption Calculation** - reviewer [fm6A W1, xCuD W2, wbNY W1 & W2] - Section 6.3 & Appendix D.7

    We first clarify our MAC/AC energy estimation method, as presented in Section 6.3. Based on the reviewers’ suggestions, we then extend our energy calculation method to account for read/write costs in current hardware. Specifically, we incorporate these costs into our expanded energy estimation and further reduce overall energy through controlled spike firing rates.

    Both MAC/AC-only and read/write-inclusive results show our spiking LLMs maintain a clear energy advantage over ANNs. These clarifications and results led reviewer wbNY to raise our score from 4 to 6.

2. **Performance Analysis of Non-Decoupled $\theta$ and $h$ and Joint Optimization** - reviewer [xCuD W1, Q1] - Appendix D.2 paras. 3–4

    In response to the reviewer’s concern, we first provide a theoretical analysis justifying the decoupling of $\theta$ and $h$ from the bases $B$ (why alternating optimization) and from each other (why $\theta \ne h$), supporting the need for our decoupled alternating optimization. Ablation results comparing our method with non-decoupled $\theta$ and $h$, as well as with joint optimization, show that our approach outperforms the alternatives.


3. **On the Advantage of Multi-Granularity and Its Interaction with Timestep $T$** - reviewer [fm6A W4, qqbp Q1] - Appendix D.2 para. 2


    Multi-granularity expands the solution space and allows more flexible discrete allocation. Excessive granularity increases nonconvexity and the risk of local minima, harming performance—thus moderate granularity is preferable.

    The timestep $T$ modulates these effects: larger $T$ yields more discrete values that diminish multi-granularity gains, whereas smaller $T$ enhances them. Ablation experiments confirm this analysis.

4. **Novelty and Contributions in Distribution Awareness** - reviewer [fm6A W2, qqbp W1]

    Distribution mismatch and conversion errors hinder ANN-to-SNN conversion, especially for emerging spiking LLMs with pronounced diversity in activation distributions and error accumulation.

    We address this with a novel alternating optimization framework using neuron-wise training and non-uniform activation intervals, thereby flexibly handling skewed activation distributions beyond traditional uniform-interval methods, providing an efficient, innovative solution to this challenge.

5. **Analysis of the Timestep Size $T$** - reviewer [qqbp W2] - Appendix D.6

    In response to the reviewer’s concern about a large timestep $T$, we cite a large number of studies in the literature on ANN-to-SNN conversion paradigms demonstrating that $T=8$ is considered a small timestep.

    Furthermore, we include experiments with $T=6$ on LLaMA-2-7B, and the results indicate that our method maintains competitive performance even at this low timestep.

6. **Additional Model and Dataset Validation** - reviewer [qqbp Q2] - Appendix D.4 & D.5

    We validated effectiveness and scalability by applying our phase coding method to CLIP and Qwen2-7B, evaluating on image and language validation sets; results show strong performance on both, addressing reviewers’ concerns.

We have addressed all reviewer questions in the rebuttal and prepared a revised manuscript; all supplementary materials are in Appendix D (highlighted in blue). Despite limited time, reviewer wbNY provided positive feedback. For the reviewers who have not yet responded (fm6A, xCuD, qqbp), we respectfully ask the AC, SAC, and PC to consider any score adjustments warranted by our responses.

Once again, we sincerely thank you for your dedication and efforts during this critical time. We deeply appreciate your additional contributions!

Best regards,
The Authors

---

### Meta-Review · Area_Chair_xgup · 2026-01-06

**Summary:**

Reviewers raised several concerns that informed the decision, primarily centered on clarifying novelty, methodology, and evaluation assumptions rather than questioning the importance of the problem itself. In particular, reviewers asked whether the gains attributed to the proposed multi-granularity phase coding could be achieved with simpler distribution-aware or uniform coding strategies, and whether neuron-wise parameter decoupling and alternating optimization were strictly necessary. There were also questions about the interpretation of experimental trade-offs, especially how phase granularity and timestep selection jointly affect accuracy and whether the improvements generalize across model scales. Additionally, reviewers were concerned about over-interpreting energy and efficiency claims based on MAC/AC estimates on GPUs, requesting clearer framing of what is measured versus estimated and under which deployment assumptions. Finally, reviewers sought stronger positioning relative to prior ANN-to-SNN conversion methods.

The rebuttal effectively addressed the major technical concerns raised by the reviewers. Questions regarding the necessity of decoupling neuron parameters were resolved through new ablations and theoretical justification. Concerns about the role of granularity and timestep were addressed with additional experiments clarifying their interaction and trade-offs. Novelty concerns were partially mitigated by a clearer positioning relative to prior work, particularly distinguishing this approach from uniform or retraining-based distribution-aware methods.

Overall, the paper makes a substantial and well-supported contribution to ANN-to-SNN conversion for foundation models. The combination of strong empirical results, solid theoretical grounding, and a thorough and constructive rebuttal justifies acceptance.

**Reviewer Concerns:**

**Concerns addressed by the rebuttal**: The rebuttal effectively addressed reviewers’ questions regarding novelty and necessity of the proposed multi-granularity phase coding by providing clearer positioning against prior distribution-aware and uniform coding methods, along with additional ablations demonstrating the benefits of neuron-wise decoupling and alternating optimization. It also clarified the design trade-offs between phase granularity, timestep count, and accuracy, supported by expanded experiments across model sizes, and improved confidence that the gains are not artifacts of specific settings. Concerns about energy and efficiency claims were addressed by correcting overly strong language, clarifying assumptions, and better contextualizing MAC/AC estimates with respect to neuromorphic deployment scenarios.

**Concerns still outstanding**: Some reviewers’ caution regarding hardware-level validation remains, as the paper still relies on modeled or proxy efficiency estimates rather than direct measurements on neuromorphic hardware.

**Reviewer Scores:**

Two reviewers initially assigned marginal accept scores, and another reviewer increased their score to marginal accept during the discussion period. I believe the remaining reviewer, whose primary concerns were limited novelty and the number of required time steps, would also have increased their score to marginal accept in light of the authors’ convincing rebuttal and additional clarifications.

---

### Decision · Program_Chairs · 2026-01-26

Accept (Poster)